# ReasonIR: Training Retrievers for Reasoning Tasks

**Rulin Shao**[*♡♠] **Rui Qiao**[*†‡] **Varsha Kishore**[♣] **Niklas Muennighoff**[◇]
**Xi Victoria Lin**[♠] **Daniela Rus**[□‡] **Bryan Kian Hsiang Low**[†‡] **Sewon Min**[♭]
**Wen-tau Yih**[♠] **Pang Wei Koh**[♡♣] **Luke Zettlemoyer**[♡♠]
[♡]University of Washington  [♠]Meta  [†]National University of Singapore
[‡]Singapore-MIT Alliance for Research and Technology
[♣]Allen Institute for Artificial Intelligence  [◇]Stanford University
[□]Massachusetts Institute of Technology  [♭]University of California, Berkeley
rulins@cs.washington.edu  rui.qiao@smart.mit.edu
🐙 Code  🤗 Model&Data

## Abstract

We present REASONIR-8B, the first retriever specifically trained for general reasoning tasks. Existing retrievers have shown limited gains on reasoning tasks, in part because existing training datasets focus on short factual queries tied to documents that straightforwardly answer them. We develop a synthetic data generation pipeline that, for each document, produces a challenging and relevant query that requires reasoning to match, as well as a plausibly related but ultimately unhelpful hard negative. By training on a mixture of this synthetic data and existing public data, REASONIR-8B achieves a new state-of-the-art of 29.9 nDCG@10 on BRIGHT, a widely-used reasoning-intensive information retrieval (IR) benchmark. In addition, REASONIR-8B uses test-time compute more effectively: on BRIGHT, its performance consistently increases with longer and more information-rich rewritten queries; it outperforms other retrievers when combined with our simple-yet-effective tie-breaking LLM reranker (36.9 nDCG@10). When applied to RAG tasks, REASONIR-8B improves MMLU and GPQA performance by 6.4% and 22.6% respectively, relative to the closed-book baseline, outperforming other retrievers and search engines. Our training recipe is general and can be easily extended to future LLMs.

## 1 Introduction

Retrieval-augmented generation (RAG) has been widely used in knowledge-seeking tasks such as factual question-answering (Asai et al., 2024; Borgeaud et al., 2022; Lewis et al., 2020; Wang et al., 2023a; Zhang et al., 2023b). In such tasks, one can often find documents that directly answer the question. However, for complex tasks that require reasoning, it is often helpful to retrieve a broader set of documents—e.g., background knowledge that contains preliminaries, tutorials that present effective reasoning patterns, or demonstration questions solved using similar methodologies. We refer to such retrieval as **reasoning-intensive retrieval**. Existing retrievers are generally trained on datasets that focus on short factual queries with straightforward matches to relevant documents, and have consequently struggled with reasoning-intensive retrieval (BehnamGhader et al., 2022; Su et al., 2024).

In this work, we present REASONIR-8B, the first bi-encoder retriever developed specifically for reasoning-intensive retrieval. The key element is REASONIR-SYNTHESIZER (§4), a recipe for synthetically generating reasoning-intensive retrieval data that we then use for contrastive training. REASONIR-SYNTHESIZER generates two types of training data: (1) *varied-length* queries and their corresponding synthesized documents, which are of diverse lengths and are designed to extend the effective context length for our retriever; and (2) *hard queries*, reasoning-intensive queries that we generate based on real seed documents. For both

---

[*]Equal contributions.

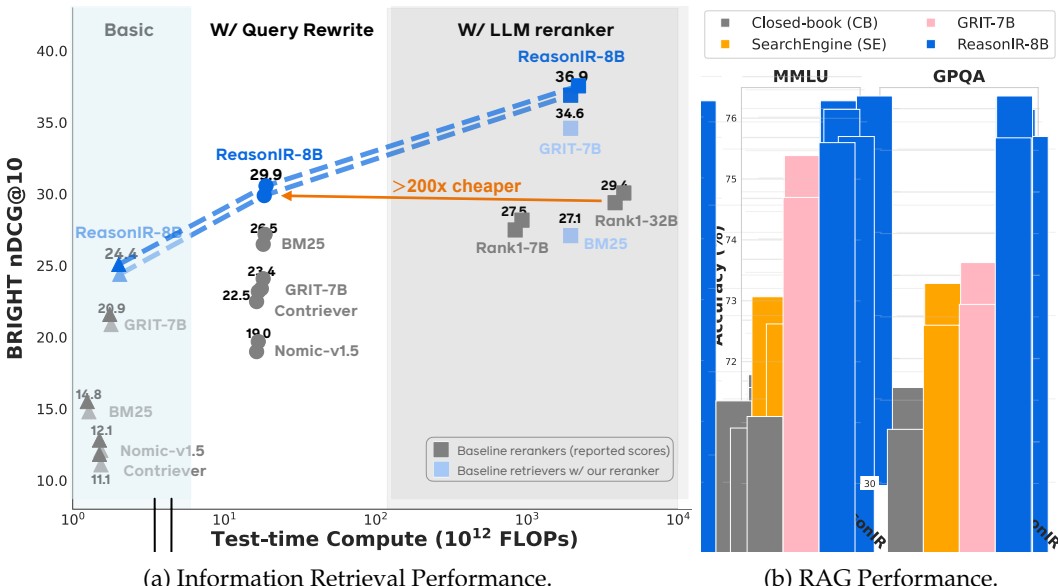

Figure 1: **(a) Performance against test-time compute on the reasoning-intensive information retrieval (IR) benchmark BRIGHT.** REASONIR-8B achieves new state-of-the-art scores, demonstrating that efficient bi-encoders can outperform significantly more expensive reranker baselines. We also introduce a simple yet effective LLM reranking method, combined with which REASONIR-8B achieves a new SOTA RBRIGHT score of 36.9 nDCG@10 (§4.6. **(b) Performance on Retrieval-augmented generation (RAG) benchmarks MMLU and GPQA.** REASONIR-8B outperforms other retriever and search engine baselines. The IR benchmark directly measures retrieval quality using annotated target documents, while RAG benchmarks measure the performance of LM responses that incorporate retrieved information.

types of queries, we also synthesize hard negatives—documents that appear superficially relevant but are actually unhelpful for the query—using a multi-turn approach, as we find that previous hard-negative-mining approaches (Luan et al., 2021) do not work well for reasoning-intensive queries. Our analysis demonstrates that the synthetic data generated by REASONIR-SYNTHESIZER is significantly more challenging and covers a broader range of query lengths compared to existing training datasets, both of which are important for improving reasoning-intensive retrieval performance.

We trained REASONIR-8B by fine-tuning LLAMA3.1-8B (Touvron et al., 2023) on a combination of public datasets and the synthetic data generated by REASONIR-SYNTHESIZER. We evaluate on both reasoning-intensive IR and RAG benchmarks (§5), as shown in Figure 1. REASONIR-8B achieves state-of-the-art results on BRIGHT (Su et al., 2024)—24.4 nDCG@10 using original queries, 29.9 with GPT4-rewritten queries, and 36.9 when further combined with an LLM reranker. Moreover, REASONIR-8B with query rewriting outperforms recent LLM reranker baselines while requiring over 200× less compute. On the reasoning-intensive RAG tasks MMLU (Hendrycks et al., 2020) and GPQA (Rein et al., 2024), where the retrieved results are fed to an LLM[1], REASONIR-8B achieves 6.39% and 22.58% relative gains respectively over closed-book baselines and outperforms other retriever and search engine baselines.

REASONIR-8B can also be effectively combined with test-time techniques. First, we studied rewriting queries to make them increasingly detailed and informative (e.g., by brainstorming about potentially useful types of information to answer the query). REASONIR-8B consistently benefits from longer rewritten queries, whereas other retrievers plateau or even worsen with longer queries; thus, REASONIR-8B enables the length of the rewritten queries as a new dimension of test-time scaling. Second, we found that REASONIR-8B

---

[1]Llama3.1-8B-Instruct and Qwen-2.5-7B-Instruct, respectively.

outperforms other retrievers when used with an LLM reranker, as it achieves a higher recall in its retrieved documents.

Our proposed method can be readily adapted to incorporate newer LLMs for synthetic data generation in REASONIR-SYNTHESIZER or as base models for training REASONIR-8B. We release our code, model, and data recipe to facilitate future research.

## 2 Preliminaries

Let $f$ denote an LM, which takes an input $x$ from a text space $\mathcal{X}$ and produces a distribution $f(x)$ from which the output $y \sim f(x)$ can be sampled. Let $\mathcal{D} = \{d_1, d_2, \ldots, d_n\}$ denote a datastore of $n$ documents. A bi-encoder retriever $h$ encodes a query $q \in \mathcal{X}$ and a document $d \in \mathcal{D}$ independently as vector embeddings $h(q)$ and $h(d)$ to produce a cosine similarity score $s(q, d) = \cos(h(q), h(d))$. Instead of outputting an answer $y$ directly to a query $q$ as $y \sim f(q)$, a RAG-based approach first retrieves the top-$k$ scored documents $D_k \subseteq \mathcal{D}$ from the datastore, then augments the original query with $D_k$ to generate more informative responses $y \sim f([q, D_k])$, where $[q, D_k] \in \mathcal{X}$ denotes the concatenation of query and documents.

Our objective is to improve the performance of RAG-powered LMs on hard, reasoning-intensive queries by improving retrieval quality, i.e., the relevance of the top-$k$ retrieved documents $D_k$. Information retrieval (IR) for such queries is more challenging because these queries exhibit low lexical and semantic overlap with relevant documents.

**Query rewriting and LLM reranking.** A reasoning-intensive query can be refined and enriched with more lexically and semantically relevant content through a query rewriter $g(\cdot; c)$ with a length configuration $c$ and chain-of-thought reasoning, producing a rewritten query $\tilde{q} = g(q; c)$, namely REASON-QUERY. The length of a REASON-QUERY is a potential test-time scaling factor that influences the retrieval quality, which we will investigate in detail in Section 3. LLM reranking is another test-time technique to further improve retrieval quality—it reaccesses the top-K retrieved documents (where $K \gg k$). An LLM assigns new relevance scores to each document, rearranging their order and selecting the k most relevant ones for final use.

**Retriever training and hard negative mining.** To train the dense retriever $h$, we use the standard contrastive learning objective (Chen et al., 2020; Gutmann & Hyvärinen, 2010):

$$\ell(q) = -\log \frac{\exp\left(\tau \cdot \cos(h(q), h(d^+))\right)}{\sum_{d_j \in \{d^+\} \cup D^-} \exp\left(\tau \cdot \cos(h(q), h(d_j))\right)} , \tag{1}$$

where $\cos(u, v) = u^\top v / \|u\| \|v\|$ is cosine similarity, $d^+ \in D^+$ is a positive document for the query $q$, $D^-$ are the negative documents for $q$, and $\tau$ is the temperature which is set to 0.02 in our experiments. Contrastive training optimizes the retriever $h$ to embed queries $q$ closer to relevant (positive) documents $D^+$ than to irrelevant (negative) ones $D^-$. Since computing distances with all $D^-$ is expensive, prior work (Robinson et al., 2020) mines **hard negatives**—irrelevant documents $\tilde{d}^- \in D^-$ for which $\cos(h(q), h(\tilde{d}^-))$ is large such that the retriever is likely to confuse as relevant—for training instead of using all negatives. The rationale is that the denominator in Equation 1 can be effectively approximated by these hard negatives. Hence, the curation of hard negatives is crucial for effective contrastive training. In this work, $D^-$ consists of both in-batch negatives and curated hard negatives.

## 3 Pilot Study: Examining Retrieval Datasets and Test-Time Scaling

We start with a pilot study to examine the limitations of existing retrievers. For evaluation, we use BRIGHT (Su et al., 2024), a widely-adopted benchmark for reasoning-intensive retrieval, spanning 12 subjects such as biology, economics, math, and coding. Following the original paper, we report the averaged nDCG@10 score across 12 subjects, which measures the ranking quality of the top-10 retrieved documents.

**Existing public training datasets are helpful for factual retrieval but not for reasoning-intensive retrieval.** We find that the queries used in existing public datasets for training retrievers are much shorter and simpler than the queries in common reasoning tasks. For example, public datasets such as Natural Questions (NQ) (Kwiatkowski et al., 2019) and MS MARCO (Nguyen et al., 2016) have average query lengths of 20 and 21 tokens, respectively. Queries from these two datasets are mostly simple factual questions, whose relevant documents can often be retrieved using direct lexical or semantic matching. However, queries in reasoning benchmarks are much longer and more complex. For example, BRIGHT has an average length of 194 tokens, and reasoning is required to retrieve positive documents for its queries. We show 2 qualitative examples from NQ and BRIGHT in Appendix L (Figure 21). This finding implies a gap between existing retriever behaviors optimized for factual retrieval and those required for reasoning-intensive tasks.

**Longer effective context length is desirable to better leverage test-time scaling through query rewriting.** We also study how query rewriting, as a popular test-time technique, can help reasoning-intensive retrieval and how we can make a retriever work better with it. We first investigate whether including more reasoning information in REASON-QUERY obtained through query rewriting can help more on BRIGHT. To study this, we adapt the query rewriting technique introduced in Section 2 to generate REASON-QUERY of different query lengths (details in Appendix B). As shown in Figure 2, all retrievers perform better when we start scaling the length of REASON-QUERY from 64 tokens to 256. However, as the length goes beyond a certain point, the improvement of dense retrievers, GRIT-7B (Muennighoff et al., 2024) and Nomic-v1.5 (Nussbaum et al., 2024), plateaus with increasing context length, while BM25 can still benefit. We note that GRIT-7B's training limit of 256 tokens for queries hinders its ability to embed long reasoning queries. While Nomic was trained to handle very long contexts ($\geq$ 10,000 tokens), this differs from typical reasoning queries (e.g., 64–2,048 tokens). These findings indicate that **the length of a rewritten query can be a new dimension of test-time scaling and a longer effective context length is desirable for long rewritten queries.**

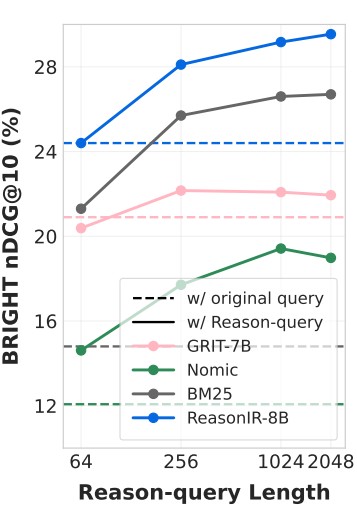

Figure 2: Query length scaling with query rewriting on BRIGHT. REASONIR-8B is our trained retriever that will be described in Section 5.

An alternative approach of utilizing test-time compute is to write and retrieve for multiple subqueries, i.e., query decomposition (Zhou et al., 2022). Query decomposition has been shown to be effective in multi-hop retrieval tasks (Wang et al., 2024; Jin et al., 2025), where the question can be easily decomposed into simple factual sub-queries. However, we evaluated a popular query decomposition method implemented by LangChain[2] on BRIGHT and found that it reduces performance from 12.1 to 10.5 with Nomic and 20.4 to 17.3 with GRIT-7B when compared with directly retrieving with the original query (detailed results in Appendix C). This indicates that **an information-rich long query is better than several decomposed short queries on BRIGHT**. Unless otherwise stated, we refer to query rewriting as the former approach, i.e., writing a long and information-rich query.

## 4 REASONIR: Synthesizing Hard and Varied-length Retriever Training Data

Our pilot study suggests two directions for improving retriever performance: training on reasoning-intensive queries and improving the effective context length of the retriever. In this section, we present REASONIR-SYNTHESIZER, a general pipeline that produces training data to improve the retriever's performance on reasoning-intensive tasks. We consider 3

---

[2]https://www.langchain.com/

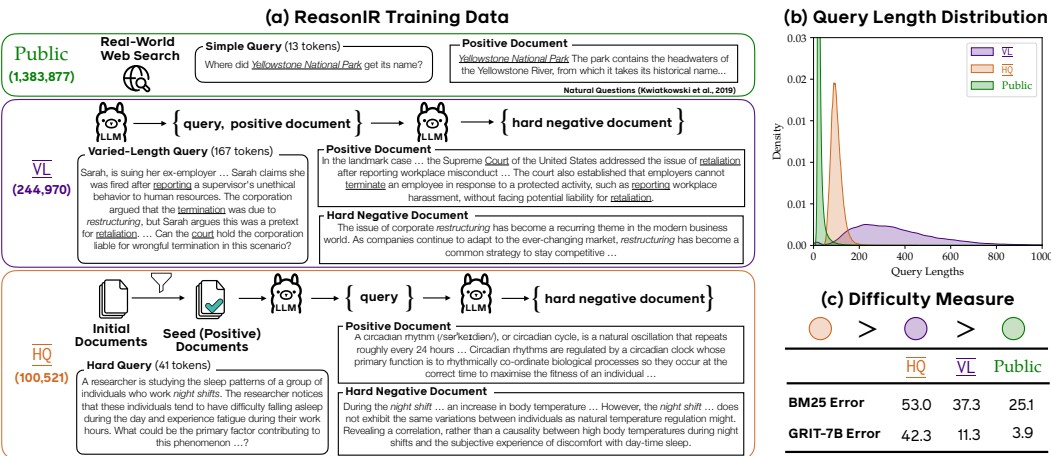

Figure 3: (a) Qualitative examples of the three types of training data used in the REASONIR training recipe and the synthetic generation pipeline, REASONIR-SYNTHESIZER, used to generate varied-length data (VL) and hard query data (HQ). (b) Query length distribution of the public, VL, and HQ data. (c) Difficulty of the public, VL, and HQ data, measured by the error rates of BM25 and GRIT-7B (i.e., how frequently they assign a higher similarity score to the hard negative document than the positive document).

types of training data: (1) public data to specifically train a general autoregressive LLM for retrieval; (2) varied-length (VL) data to extend the effective context length of the retriever for input queries; and (3) hard query (HQ) data to improve the retriever's ability to handle reasoning-intensive queries. The pipeline and data statistics are shown in Figure 3.

## 4.1 Public Data

Following Muennighoff et al. (2024), we include a set of public training data collected from real-world applications, such as MS MARCO (Nguyen et al., 2016), Natural Questions (Kwiatkowski et al., 2019), HotpotQA (Yang et al., 2018) (details in Appendix D). These datasets provide a diverse coverage of topics and languages for adapting an autoregressive LLM to embedding tasks that focus on short factual queries.

## 4.2 Varied-length Synthetic Query and Positive Document Generation (VL)

To enable a longer effective context length for the query inputs, we first generate long queries (300–2,000 natural words) to encourage the retriever to leverage rich information from lengthy and complex queries. For simplicity, we ask the LLM to also generate a positive document for the query, following the distillation idea in Wang et al. (2023b). The queries generated cover a wider range of lengths, so we name it "varied-length" (VL) data. We present the generation template in Appendix A.3.

## 4.3 Reasoning-intensive Document-to-query Generation (HQ)

Previous work has shown that LLM-generated reasoning-intensive questions often lack the requisite diversity and difficulty, and they often involve human-in-the-loop to generate high-quality difficult questions (Shah et al., 2024; Chiu et al., 2024). To improve diversity and eliminate the need for human effort, we synthesize reasoning-intensive training data by generating hard queries (HQ) from high-quality documents using a "human-like brainstorm guideline" for hard query generation.

**Reasoning-worthy seed document selection.** We define a reasoning-worthy document as one that contains knowledge that can potentially aid in understanding and solving

reasoning tasks. We assume that such documents are more helpful for reasoning-intensive query generation. In contrast, when working with less informative sources such as subjective Web forum comments or sparse event descriptions, LLMs typically struggle to generate challenging questions. The documents collected by Su et al. (2024) cover a diverse range of scientific domains, such as biology, economics, mathematics, and coding, and many of them have been cited in human answers to reasoning-intensive questions on forums. Therefore, we use these documents as the initial knowledge pool and further apply the FineWeb-Edu classifier (Penedo et al., 2024) to score each document based on its educational value. We remove those with scores lower than 2, which usually contain gibberish or subjective content, and use the filtered documents as seed documents for hard query generation.

**Reasoning-intensive document-to-query generation.** An ideal set of reasoning-intensive queries has three properties: (1) challenging — demanding reasoning beyond simple lexical or superficial semantic matching; (2) self-contained — understandable without the presence of the seed document; (3) diverse — imitating diverse question styles in various problem-solving scenarios. As previous work has shown unsuccessful attempts on directly prompting an LLM to generate difficult questions (Shah et al., 2024), we develop a new document-to-query generation method—we provide the LLM with a reasoning-worthy document and then instruct it to come up with hard queries following a human-like brainstorming guideline: specifically, we ask the LLM to reason about the background knowledge, common problem-solving patterns, and realistic scenarios before formulating a difficult question. We also instruct the model to avoid document-dependent questions that reference specific terms from the seed document, as they are not self-contained without the document's presence (full instructions in Appendix A.1 and qualitative examples in Appendix L). The seed document also serves as the positive document. Our design of reasoning-intensive question generation is general, so it can also be used for various synthetic data generation tasks beyond reasoning-intensive retrieval.

## 4.4 Multi-turn Hard Negative Generation

In addition to creating queries and positive documents, finding hard negatives that are sufficiently difficult (also called "hard negative mining") has been shown to be important to the success of contrastive training (Kalantidis et al., 2020; Robinson et al., 2020). Existing research typically identifies hard negatives by selecting top-ranked but irrelevant documents from a retriever such as BM25 (Luan et al., 2021). However, we find that this does not work for reasoning-intensive queries for 3 reasons: First, existing retrievers perform poorly on reasoning-intensive queries (Su et al., 2024), making them inadequate for hard negative mining. Second, the goal of retrieval has shifted from finding documents that contain direct answers to finding a wide range of documents that are helpful for reasoning, which increases the risk of mining incorrect hard negatives from the same knowledge source. Third, the seed document may not be the most relevant to the generated query since the query is created without knowledge of the entire datastore, potentially causing false negatives that harm training performance if used. Therefore, we propose to directly generate new hard negatives for reasoning-intensive queries. We find that prompting the LLM to generate queries and hard negatives simultaneously often results in short and easy negatives. We resolve this issue by generating the hard negative in a separate turn, conditioning on the previously obtained query and positive document for both $\overline{VL}$ and $\overline{HQ}$ data (detailed instructions in A.2).

## 4.5 Difficulty and Length Analysis

We randomly sample 100,000 examples from the public data, $\overline{VL}$, and $\overline{HQ}$, respectively, for difficulty and length analysis. We visualize the length distribution of the 3 types of data in Figure 3(b), showing that $\overline{VL}$ spans a significantly longer range of query lengths than the other 2 types of data. The queries of $\overline{HQ}$ are shorter than $\overline{VL}$ but longer than the public data. We also compute the error rates (the frequency that a retriever gives the hard negative a higher similarity score than the positive document) of BM25 and GRIT-7B on these data types. We assume that BM25 error rates correspond to the difficulty of the data to be solved using naive lexical matching. GRIT-7B error rates measure the difficulty of the

examples being solved by a more advanced retriever that is good at semantic matching tasks. Figure 3(c) shows that HQ is significantly more challenging and that the public data can be mostly solved using a retriever that is good at semantic matching.

### 4.6 REASONIR-Rerank: A Simple but Effective Tie-Breaking LLM Reranking Method

Retrieve-then-rerank is a common practice for better retrieval performance, where LLM rerankers have been shown to be effective on reasoning-intensive retrieval (Su et al., 2024). Naive LLM reranker (Sun et al., 2023a;b) uses an off-the-shelf LLM to give an integer helpfulness score within a range (e.g., 0–5). Recent works (Weller et al., 2025; Zhuang et al., 2025) found that naive LLM rerankers perform poorly on reasoning-intensive retrieval and thus developed better LLM rerankers for reasoning-intensive retrieval by distilling from the reasoning traces for reranking outputs produced by a large reasoning model, such as DeepSeek-R1 (Guo et al., 2025). However, the costs of creating such training data and performing inference-time reasoning are both high when using these methods. It is desirable to have an LLM reranker that costs similarly to the naive LLM reranker while matching or even outperforming the reasoning-based LLM rerankers. To solve this, we first investigate the reason that causes the naive LLM reranker to have poor performance and find that it is mainly because naive LLM rerankers result in too many ties in their reranking results (Appendix G.1). To resolve this, we propose to interpolate the reranking scores with the scores given by the base retriever, named REASONIR-Rerank. In this way, we find the interpolation can effectively break the ties and result in even higher performance than existing reasoning-based reranker baselines on BRIGHT (Section 5.1). In the remainder of the paper, we refer to REASONIR-Rerank using QWEN2.5-32B-INSTRUCT as QWENRERANK for convenience.

## 5 Experiments

**General setup.** We use LLAMA3.1-70B-INSTRUCT for synthetic data generation and train a bi-encoder retriever, REASONIR-8B, using LLAMA3.1-8B as the base model. Our training data include 1,383,877 public training samples, 244,970 VL samples, and 100,521 HQ samples. To enhance the quality of the embedding, we modify the attention mask of LLAMA3.1-8B from a causal attention mask to a bi-directional attention mask (Muennighoff et al., 2024). We evaluate our model on both IR and RAG tasks. We use BRIGHT (Su et al., 2024) for IR evaluation and MMLU (Hendrycks et al., 2020) and GPQA (Rein et al., 2024) with 3 in-context retrieved passages for RAG evaluation with LLAMA3.1-8B and QWEN2.5-7B-INSTRUCT as the reader models, respectively. For query rewriting experiments, we use the reader model itself to rewrite queries. Details of the training and evaluation setup can be found in Appendix E.

### 5.1 Reasoning-intensive Information Retrieval (IR) Performance

We compare our model with existing bi-encoder retrievers and LLM rerankers. Note that the LLM rerankers we used are cross-encoder models that output a similarity score for each query-document pair by processing them within a single context as input. They are also much more expensive at test-time, as analyzed in Section 5.4. For bi-encoder baselines, we include the sparse retriever BM25 (Robertson et al., 2009) and dense retrievers E5 (Wang et al., 2023b) and GRITLM-7B (Muennighoff et al., 2024). For LLM reranker baselines, we compare with RankLLaMA-7B (Ma et al., 2024) and Rank1-7B (Weller et al., 2025), which rerank the top-100 documents retrieved by BM25 using GPT-4 REASON-QUERY.

**REASONIR-8B achieves SOTA scores on BRIGHT.** As shown in Figure 1 and Table 1, REASONIR-8B outperforms existing retrievers and expensive LLM rerankers on both original queries and REASON-QUERY, achieving nDCG@10 scores of 24.4 and 29.9, respectively. Notably, REASONIR-8B outperforms LLM Rerankers using 200× less compute (§5.4). Additionally, we demonstrate that REASONIR-8B can still perform well with a small-sized query rewriter, LLAMA3.1-8B-INSTRUCT, achieving an nDCG@10 score of 28.0 on BRIGHT.

| | StackExchange | | | | | | | Coding | | Theorem-based | | | Avg. |
|---|---|---|---|---|---|---|---|---|---|---|---|---|---|
| | Bio. | Earth. | Econ. | Psy. | Rob. | Stack. | Sus. | Leet. | Pony | AoPS | TheoQ. | TheoT. | |
| *Evaluate with original query* | | | | | | | | | | | | | |
| BM25 | 19.2 | 27.1 | 14.9 | 12.5 | 13.5 | 16.5 | 15.2 | 24.4 | 7.9 | 6.0 | 13.0 | 6.9 | 14.8 |
| Contriever | 9.2 | 13.6 | 10.5 | 12.1 | 9.5 | 9.6 | 8.9 | 24.5 | 14.7 | 7.2 | 10.4 | 3.2 | 11.1 |
| GritLM-7B | 25.0 | 32.8 | 19.0 | 19.9 | 17.3 | 11.6 | 18.0 | 29.8 | **22.0** | 8.8 | 25.1 | 21.1 | 20.9 |
| **REASONIR-8B** | **26.2** | 31.4 | **23.3** | **30.0** | 18.0 | **23.9** | 20.5 | **35.0** | 10.5 | **14.7** | **31.9** | **27.2** | **24.4** |
| OpenAI | 23.7 | 26.3 | 20.0 | 27.5 | 12.9 | 12.5 | 20.3 | 23.6 | 2.5 | 8.5 | 23.8 | 12.3 | 17.8 |
| Voyage | 23.6 | 25.1 | 19.8 | 24.8 | 11.2 | 15.0 | 15.6 | 30.6 | 1.5 | 7.4 | 26.1 | 11.1 | 17.7 |
| Google | 23.0 | **34.4** | 19.5 | 27.9 | 16.0 | 17.9 | 17.3 | 29.6 | 3.6 | 9.3 | 21.5 | 14.3 | 19.5 |
| *Evaluate with* LLAMA3.1-8B-INSTRUCT REASON-QUERY | | | | | | | | | | | | | |
| **REASONIR-8B** | 37.8 | 39.6 | 29.6 | 35.3 | 24.1 | 31.1 | 27.4 | 28.8 | 14.5 | 9.2 | 26.6 | 32.3 | 28.0 |
| **+ BM25 (Hybrid)** | 51.9 | 50.6 | 24.0 | 40.6 | 26.9 | 31.0 | 28.5 | 26.2 | 17.8 | 9.2 | 22.3 | 22.5 | 29.3 |
| *Evaluate with* GPT4 REASON-QUERY | | | | | | | | | | | | | |
| Contriever | 37.5 | 40.5 | 22.6 | 27.1 | 15.2 | 22.6 | 19.6 | 22.5 | 13.8 | 8.1 | 24.1 | 16.2 | 22.5 |
| GritLM-7B | 33.2 | 33.0 | 23.3 | 30.6 | 15.2 | 17.5 | 21.7 | **33.2** | 11.7 | 6.8 | 26.9 | 28.0 | 23.4 |
| BM25 | 53.6 | 53.6 | 24.3 | 38.6 | 18.8 | 22.7 | 25.9 | 19.3 | 17.7 | 3.9 | 20.2 | 18.9 | 26.5 |
| + RankLLaMA-7B | 17.5 | 15.5 | 13.1 | 13.6 | 17.9 | 6.9 | 16.9 | 8.4 | **46.8** | 2.2 | 4.5 | 3.5 | 13.9 |
| + Rank1-7B | 48.8 | 36.7 | 20.8 | 35.0 | 22.0 | 18.7 | **36.2** | 12.7 | 31.2 | 6.3 | 23.7 | 37.8 | 27.5 |
| + Rank1-32B | 49.7 | 35.8 | 22.0 | 37.5 | 22.5 | 21.7 | 35.0 | 18.8 | 32.5 | **10.8** | 22.9 | 43.7 | 29.4 |
| **REASONIR-8B** | 43.6 | 42.9 | **32.7** | 38.8 | 20.9 | 25.8 | 27.5 | 31.5 | 19.6 | 7.4 | 33.1 | 35.7 | **29.9** |
| **+ BM25 (Hybrid)** | 55.9 | **54.9** | 29.6 | 42.9 | 23.0 | 27.9 | 29.8 | 27.9 | 25.8 | 7.2 | 33.7 | 25.8 | **32.0** |
| **+ QWENRERANK** | **58.2** | 53.2 | 32.0 | **43.6** | **28.8** | **37.6** | 36.0 | **33.2** | 34.8 | 7.9 | 32.6 | **45.0** | **36.9** |

Table 1: Reasoning-intensive information retrieval performance on BRIGHT. We highlight proprietary models with a gray background. "+ BM25 (Hybrid)" refers to a hybrid version of REASONIR-8B, where we combine the similarity scores from REASONIR-8B and BM25 by interpolating them with a ratio of 0.5. "QWENRERANK" is our proposed simple-yet-effective hybrid reranker method built on QWEN2.5-32B-INSTRUCT that does not require any training or long reasoning outputs (more details in Section 4.6).

**REASONIR-8B benefits from test-time scaling with query rewriting.** As shown in Figure 2 in our pilot study, REASONIR-8B continues to benefit from longer queries with query rewriting, while other dense retrievers (GRIT-7B and nomic-v1.5-text) have diminishing gains or even decreased performance when scaling up the query length. This indicates that our retriever can leverage the rich information in the long rewritten queries better than existing retrievers.

**REASONIR-8B can form an ensemble with a sparse retriever or be combined with an LLM-based reranker for better retrieval.** Interpolating the retrieval scores of REASONIR-8B and BM25 (with a ratio of 0.5) further improves nDCG@10 to 32.0. Our retriever can also be combined with an LLM-based reranker. In Table 1, we show REASONIR-8B achieves an nDCG@10 score of 36.8 on BRIGHT when combined with QWENRERANK, which is a simple zero-shot QWEN2.5-32B-INSTRUCT reranker (detailed prompts in Appendix E.2). Our QwenRerank model is more performant, faster and simpler when compared with Rank1 (Weller et al., 2025), which fine-tunes QWEN-series models with reasoning traces distilled from DeepSeek-R1 (Guo et al., 2025) and generates about 300 reasoning tokens when reranking each query-document pair. We provide more results of our QWENRERANK with different baseline retrievers in Appendix G.

## 5.2 Reasoning-intensive Retrieval-augmented Generation (RAG) Performance

In this section, we evaluate using REASONIR-8B for an RAG pipeline on two reasoning-intensive open-domain benchmarks, MMLU and GPQA. For the datastore, we use MassiveDS (Shao et al., 2024) as it has been shown to be helpful for many RAG tasks. Due to computational constraints, we use a filtered version of MassiveDS, as detailed in Appendix E.3. We use LLAMA3.1-8B for MMLU and QWEN2.5-7B-INSTRUCT for GPQA as both the reader and query rewriter. We compare REASONIR-8B against GRIT-7B (Muennighoff et al., 2024), a SOTA retriever at a similar scale of REASONIR-8B. In addition, we also consider you.com search API (https://you.com) as a black-box search engine baseline. Due to budget limit, we run this search baseline on GPQA only.

| | | | StackExchange | | | | | Coding | | Theorem-based | | | Avg. |
|---|---|---|---|---|---|---|---|---|---|---|---|---|---|
| | Bio. | Earth. | Econ. | Psy. | Rob. | Stack. | Sus. | Leet. | Pony | AoPS | TheoQ. | TheoT. | |
| *Evaluate with original query* | | | | | | | | | | | | | |
| LLAMA3.1-8B | 12.5 | 6.5 | 7.7 | 7.7 | 3.9 | 7.5 | 8.6 | 22.0 | 17.1 | 10.5 | 7.4 | 2.0 | 9.5 |
| Public | 21.4 | 30.3 | 17.8 | 24.7 | 18.6 | 18.8 | 18.8 | 30.0 | 6.7 | 12.1 | 21.4 | 15.2 | 19.6 |
| Public+HQ | 21.0 | 31.3 | 18.4 | 25.1 | 15.7 | 18.4 | 14.3 | 34.1 | 5.2 | 9.5 | 33.7 | 24.4 | 20.9 |
| Public+VL | 28.4 | 35.8 | 22.5 | 28.4 | 18.4 | 19.5 | 18.7 | 34.5 | 12.3 | 11.4 | 24.4 | 23.6 | 23.2 |
| Public+EQVL | 26.8 | 33.8 | 23.4 | 30.1 | 21.1 | 21.9 | 21.5 | 31.0 | 6.5 | 10.1 | 20.9 | 20.2 | 22.3 |
| Public+HQVL | 26.2 | 31.4 | 23.3 | 30.0 | 18.0 | 23.9 | 20.5 | 35.0 | 10.5 | 14.7 | 31.9 | 27.2 | **24.4** |
| *Evaluate with GPT4 REASON-QUERY* | | | | | | | | | | | | | |
| LLAMA3.1-8B | 41.3 | 25.1 | 16.8 | 17.3 | 8.7 | 10.7 | 15.7 | 6.8 | 32.3 | 0.9 | 12.3 | 4.0 | 16.0 |
| Public | 40.3 | 42.1 | 26.0 | 37.7 | 20.8 | 22.6 | 22.7 | 32.3 | 13.5 | 7.0 | 29.5 | 30.4 | 27.1 |
| Public+HQ | 37.4 | 42.7 | 26.8 | 35.3 | 18.2 | 22.1 | 20.0 | 35.0 | 14.7 | 6.7 | 34.1 | 32.7 | 27.1 |
| Public+VL | 33.8 | 41.3 | 28.9 | 40.2 | 20.6 | 24.2 | 25.9 | 34.7 | 19.6 | 4.8 | 32.5 | 29.2 | 28.0 |
| Public+EQVL | 37.9 | 42.1 | 30.6 | 40.0 | 22.1 | 25.6 | 27.4 | 31.8 | 15.5 | 6.1 | 27.3 | 28.7 | 27.9 |
| Public+HQVL | 43.6 | 42.9 | 32.7 | 38.8 | 20.9 | 25.8 | 27.5 | 31.5 | 19.6 | 7.4 | 33.1 | 35.7 | **29.9** |

Table 2: Ablation on the training data—nDCG@10 BRIGHT performance with models trained on different data sources. Each model is trained for 1000 steps using LLAMA3.1-8B as the initial checkpoint. "Public" stands for data from existing training datasets. HQ, VL, and EQ refer to hard queries, varied-length data, and easy queries, respectively.

**REASONIR-8B outperforms all baselines on MMLU and GPQA.** As shown in Figure 1(b), using REASONIR-8B for RAG improves the performance of the base LLM (closed-book) by 3.9 and 7.1 absolute points on MMLU and GPQA, respectively, outperforming the previous SOTA retriever GRIT-7B. For open-domain knowledge-seeking tasks, search engines are considered a powerful baseline because they usually have access to a more expansive knowledge base compared to in-house retrieval systems. However, in Figure 1(b), we show that RAG with you.com search underperforms REASONIR-8B on GPQA, highlighting the benefits of developing high-quality in-house datastores with a strong retriever like REASONIR-8B for reasoning-intensive RAG tasks.

**The effect of query rewriting on MMLU and GPQA.** In realistic RAG setups, a query rewriter should not be stronger than the reader model. Therefore, we use the same reader model to rewrite queries to evaluate the effect of query rewriting, as shown in Table 11. We find that applying query rewriting generally helps improve the MMLU performance—we observed 0.8, and 0.6 absolute improvements for GRIT-7B and REASONIR-8B, respectively. It also helps improve the GPQA performance of the search engine from 33.8 to 36.4. However, rewritten queries decrease all dense retrievers' performance on GPQA, which we hypothesize is caused by the small-scale reader model not being able to write good queries for this task, unlike in BRIGHT where a strong model, GPT-4, was used to write good queries. As it is out of the scope of our work, we leave it for future work to develop a better query rewriter.

## 5.3 Ablation Studies

**Ablation on Data Composition.** To study the impact of data composition, we train different retrievers using different mixes of training data (introduced in Section 4) evaluate on BRIGHT. The Public split is always included in the training set. The results are shown in Table 2. We find that the retriever trained with public data and VL outperforms the retriever trained with public data and HQ. We hypothesize that this is because HQ contains only hard queries that are longer and more challenging than those in public data, making it difficult for the model to learn effectively without the inclusion of VL during training, which helps bridge the gap in query difficulty and length. When training with both VL and HQ, the resulting retriever achieves the best performance on both the original query and REASON-QUERY, significantly outperforming any single-sourced training. Specifically, combining HQ and VL achieves an nDCG@10 score of 24.4 on original queries and 29.9 on REASON-QUERY. Our results demonstrate the synergy between VL and HQ and the necessity of including a diverse mix of synthetic data in the training pool to obtain good performance.

**Ablation on Query Difficulty.** To study the impact of query difficulty, we introduce a new type of data EQ (short for "easy query"). EQ is generated with a baseline prompt modified from Chaudhary et al. (2023). We use the same reasoning-worthy document selection and multi-turn hard-negative generation to only ablate the impact of our reasoning-intensive document-to-query generation pipeline. As shown in Table 2, adding EQ does not provide any performance improvement on top of VL, indicating that **the improved performance from adding HQ can be attributed to the generation of hard and reasoning-intensive queries**, while naively generating simple questions using in-domain documents does not help on BRIGHT.

## 5.4 Computational Analysis

We present test-time compute estimation for bi-encoders and LLM rerankers. For simplicity, we approximate the encoding/decoding cost for a token with $2N$ FLOPs, where $N$ is the number of non-embedding LLM parameters (Kaplan et al., 2020). We do not distinguish between decoding and encoding in FLOPS computation but note that auto-regressive token decoding (e.g., by LLM rerankers) often takes more wall-clock time than token embedding (e.g., by bi-encoders). Figure 1(a) shows estimated test-time FLOPs for 3 retrieval scenarios, distinguished by whether query rewriting or an LLM reranker is used. Detailed derivations and computational comparisons are in Appendix J. We find that **our bi-encoder retriever REASONIR-8B outperforms expensive LLM reranker Rank1-32B with over $200\times$ less test-time compute** (Figure 1(a)). In addition, our QWENRERANK consumes less test-time compute than Rank1-32B while being more performant.

## 6 Related Work

**Retrieval and Reasoning.** Retrieval has traditionally been regarded as less effective for reasoning-intensive tasks (BehnamGhader et al., 2022). Recent advances in reasoning models have shown significant performance improvements by integrating retrieval into their reasoning processes (Jin et al., 2025; Song et al., 2025; Wang et al., 2025). More discussion on test-time scaling for RAG systems can be found in Appendix K. However, these systems like R1-Searcher (Li et al., 2025) rely on retrieval models such as BGE (Xiao et al., 2024b), which perform well on semantic retrieval benchmarks but struggle with reasoning-heavy tasks (Su et al., 2024). At the same time, challenging benchmarks for reasoning-intensive retrieval have emerged (Xiao et al., 2024a; Wang et al., 2024; Su et al., 2024; Enevoldsen et al., 2025), yet high-performing models remain scarce.

**Synthetic data generation.** Large language models (LLMs) can be effectively prompted to generate synthetic training data (Hsieh et al., 2023; Wang et al., 2022). In information retrieval, one family of approaches leverages LLMs to synthesize simple, relevant, and task-specific queries from documents based on few-shot exemplars (Bonifacio et al., 2022; Chaudhary et al., 2023; Dai et al., 2022). Conversely, the documents can be synthesized from queries (Weller et al., 2024). Wang et al. (2023b) developed a comprehensive framework that synthesizes tasks, queries, and documents at once. However, none of these approaches prioritize reasoning-intensive retrieval.

## 7 Discussion

In this work, we explore synthetic data generation for training retrievers for reasoning tasks. Our bi-encoder retriever, REASONIR-8B, achieves significant improvements on both IR and RAG reasoning tasks. There are many potential directions for future work: for example, studying the scaling trends of such synthetic data, designing better reasoning-worthy seed document selection methods, extending it to multilingual and multimodal versions, and combining it with multi-turn reasoning models for more complex tasks.

## Acknowledgments

This work was done while RS was part of the UW-Meta AI Mentorship program; it was partially done while Rui Qiao was a visiting scholar at UW. We thank Xueguang Ma and Tong Chen for their insightful discussions. We thank Scott Geng and Xilun Chen for proofreading. We thank Koda, our lab dog, for emotional support. This work is supported by the Singapore National Research Foundation and the National AI Group in the Singapore Ministry of Digital Development and Information under the AI Visiting Professorship Programme (award number AIVP-2024-001), and by the AI2050 program at Schmidt Sciences. This research is supported by the National Research Foundation (NRF), the Prime Minister's Office, Singapore, under its Campus for Research Excellence and Technological Enterprise (CREATE) program. The Mens, Manus, and Machina (M3S) is an interdisciplinary research group (IRG) of the Singapore MIT Alliance for Research and Technology (SMART) centre.

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

## Appendix

---

**System Prompt for Reasoning-intensive Document-to-query Data Generation**

```
# Context
You are tasked with generating {num_questions} reasoning-intensive questions with
scenarios based on a given document. These questions must be standalone
(meaningful without the document) while being answerable using information from
the document as supporting evidence. The questions should specifically engage
with core concepts and principles from the document's domain.

# Question Requirements
1. Each question MUST:
- Present a complete scenario or context within itself
- Be answerable through logical reasoning and critical thinking
- Remain valid and meaningful even if the source document didn't exist
- Target higher-order thinking skills (analysis, evaluation, synthesis)
- Be domain-relevant but not document-specific
- Incorporate key concepts, terminology, and principles from the document's field
- Challenge understanding of domain-specific problem-solving approaches

2. Each question MUST NOT:
- Directly reference the document or its contents
- Be answerable through simple fact recall
- Require specific knowledge only found in the document
- Be a reading comprehension question
- Stray from the core subject matter of the document's domain

# Domain Alignment Guidelines
Before generating questions:
1. Identify the primary domain (e.g., programming, medicine, economics)
2. Extract key concepts and principles from the document
3. List common problem-solving patterns in this domain

When crafting questions:
1. Frame scenarios using domain-specific contexts
2. Incorporate relevant technical terminology naturally
3. Focus on problem-solving approaches typical to the field
4. Connect theoretical concepts to practical applications within the domain

After generating the questions step by step, reformat all questions including the
corresponding scenarios in JSON with key "hard_query":
```json
{{
    "hard_query": [ Q1, Q2, Q3, ...]
}}
```
```

Figure 4: System prompt for reasoning-intensive document-to-query data generation (HQ)

## A   Prompts

### A.1   Prompt for Reasoning-intensive Document-to-query Data Generation (HQ)

We document the system prompt and the user prompt used for HQ in Figure 4 and Figure 5.

### A.2   Prompt for Multi-turn Hard-negative Generation

We show the prompt for multi-turn hard-negative generation in Figure 6.

---

**User prompt for reasoning-intensive document-to-query data generation**

```
The document is given below:

<document>
{document}
</document>

Please start generating the questions.
```

Figure 5: User prompt for reasoning-intensive document-to-query data generation (HQ)

---

**System prompt for multi-turn hard-negative generation**

```
You have been assigned a passage generation task:

You will be provided an incomplete data with the below information
- "input": a string, a random input specified by one task.
- "positive_document": a string, a relevant document for the "input" according to
the task.

Your task is to generate a "hard_negative_document" in a JSON format:
- The "hard_negative_document" contains some relevant information with superficial
lexical overlapping, but it should be not helpful to address the question in the
input and is less relevant to the input compared with the "positive_document".

Please adhere to the following guidelines:
- The values of "hard_negative_document" should be in English.
- The "hard_negative_document" should be long documents (at least 300 words),
avoid substantial word overlaps, otherwise the task would be too easy.
- The "input", "positive_document", and "hard_negative_document" should be
independent of each other.

Your output must always be a JSON object only, do not explain yourself or output
anything else. Be creative!

Now process the below data following the above instruction:
'input': \{query\}
'positive_document': \{positive_document\}

Your response:
```

Figure 6: System prompt for multi-turn hard-negative generation

---

**System prompt for instruction generation in VL.**

```
Brainstorm a list of text matching tasks where the queries are long documents.

Here are a few examples:
- Given a document that supports a debatable argument, find another document that
contains opposite arguments.
- Provided a lengthy business proposal, retrieve competitive business strategies
in the same industry.
- Provided a stackexchange lengthy question, retrieve relevant STEM knowledge
from scientific papers.
- Given a reasoning-intensive math or coding question, retrieve demonstrations
from the textbooks that can help answer the questions.

Your output must always be a python list of strings only, with about 20 elements,
and each element corresponds to a distinct task in one sentence. Do not explain
yourself or output anything else. Be creative!
```

Figure 7: System prompt for instruction generation in VL.

---

**System prompt for instruction generation in VL.**

```
You have been assigned a text matching task: {instruction}

Your mission is to write one example for this task in JSON format. The JSON object
must contain the following keys:
- "input": a string, a random input specified by the task.
- "positive_document": a string, a relevant document for the "input" according to
the task.

Please adhere to the following guidelines:
- The values of all fields should be in English.
- Both the "input" and "positive_document" should be long documents (at least
{length} words), avoid substantial word overlaps, otherwise the task would be too
easy.
- The "input" and "positive_document" should be independent of each other.

Your output must always be a JSON object only, do not explain yourself or output
anything else. Be creative!
```

Figure 8: System prompt for final data generation in VL. We sample "length" from below 300, 300, 500, 1000, 1500, and 2000.

## A.3 Prompt for Varied Length Data Generation (VL)

For varied-length data generation, we first prompt an LLM to brainstorm a list of instructions that define potential scenarios using the prompt in Figure 7. We then prompt the LLM to further generate the query and positive document using the prompt in Figure 8. Finally, we concatenate the task instruction with the generated query as the new query and provide the new query and positive document to the LLM to further generate the hard negative document using the prompt in Figure 6.

## A.4 Prompt for Reasoning Query Rewritting REASON-QUERY

We document the system prompt and the user prompt used for rewriting queries into REASON-QUERY in Figure 9 and Figure 10. These two prompts are developed by BRIGHT (Su

---

**System prompt for reasoning query augmentation generation**

```
You are a helpful assistant
```

---

Figure 9: System prompt for reasoning query augmentation generation ($\overline{RQ}$) (Su et al., 2024).

---

**User prompt for reasoning query augmentation generation**

```
{cur_post}

Instructions:
1. Identify the essential problem.
2. Think step by step to reason and describe what information could be relevant
and helpful to address the questions in detail.
3. Draft an answer with as many thoughts as you have.
```

---

Figure 10: User prompt for reasoning query augmentation generation ($\overline{RQ}$) (Su et al., 2024).

et al., 2024) and we reuse them to ensure consistency. The length-controlled version of the prompt used for studying test-time scaling is shown in Figure 11.

### A.5 Prompts for Easy Query Generation (EQ)

We document the system prompt for the easy-query generation baseline $\overline{EQ}$ in Figure 12. We reuse the user prompt with the seed document in Figure 5.

### A.6 Prompts for Reranking

We document the prompts used to rerank top-100 candidates by using our proposed simplified Qwen-32b reranker. Figure 13 shows the prompt used for the StackExchange tasks in BRIGHT. Following Weller et al. (2025), we use data-specific prompts for the non-StackExchange tasks (see Figure 14, 15, 16, 17, 18).

### A.7 Prompts for Query Decomposition

We directly apply the query decomposition method implemented by LangChain. Figure 19 shows the prompt it used for query decomposition.

---

**User prompt for reasoning query augmentation generation with length control**

```
{cur_post}

Instructions:
1. Identify the essential problem.
2. Think step by step to reason and describe what information could be relevant
and helpful to address the questions in detail.
3. Draft an answer with as many thoughts as you have.
Your answer must be written within {MAX_TOKENS} tokens.
```

---

Figure 11: User prompt for reasoning query augmentation generation with length control.

---

**System prompt for easy query generation**

```
Given a document, generate {num_questions} questions for which the document is
relevant and useful to provide the answer. Format the generated questions in JSON
with key "questions":
```json
{{
    "questions": [ "question 1", ...]
}}
```
```

Figure 12: System prompt for easy query generation.

---

**Prompt for reranking StackExchange**

```
A document is relevant if it contains information that helps answer or address the
query.
A document is not relevant if it doesn't contain information that helps answer the
query, even if it mentions similar topics.
Is the document below relevant to answering the query below?
The answer should be 'Relevance score: X.' where X is a number from 0-5.
0 means completely irrelevant, 5 means highly relevant and completely addresses
the query. Don't output anything else.
Here is the query:
<start_query>
{}
<end_query>
Here is the document:
<start_document>
{}
<end_document>
```

Figure 13: Prompt for reranking StackExchange tasks.

---

**Prompt for reranking AoPS**

```
We want to find different but similar math problems to the following problem:
{}
A document is relevant if it uses the same class of functions and shares **any**
overlapping techniques.
Document: {}
Score the document above. The answer should be 'Relevance score: X.' where X is a
number from 0-5.
0 means completely irrelevant, 5 means highly relevant and completely addresses
the query. Don't output anything else.
```

Figure 14: Prompt for reranking AoPS.

---

**Prompt for reranking Leetcode**

```
I am looking to find different problems that share similar data structures
(of any kind) or algorithms (e.g. DFS, DP, sorting, traversals, etc.). I am
looking for problems that share one or both of these similarities to
this:
{}
Does the passage below share any similarities? e.g. if there was a textbook on
leetcode problems, this would be in the same book even though it could be in a
different chapter.
Passage: {}
Please rate the passage above. The answer should be 'Relevance score: X.' where X
is a number from 0-5.
0 means completely irrelevant, 5 means highly relevant and completely addresses
the query. Don't output anything else.
```

Figure 15: Prompt for reranking Leetcode.

---

**Prompt for reranking Pony**

```
I will use the programming language pony.
Problem: {}
But to solve the problem above, I need to know things about pony. A passage is
relevant if it contains docs that match any part (even basic parts) of the code I
will have to write for the above program.
Passage: {}
Please rate the passage above. The answer should be 'Relevance score: X.' where X
is a number from 0-5.
0 means completely irrelevant, 5 means highly relevant and completely addresses
the query. Don't output anything else.
```

Figure 16: Prompt for reranking Pony.

---

**Prompt for reranking TheoQ**

```
We want to find a document which uses the same mathematical process as this one:
{}
A document is relevant if it uses the same mathematical process as the query.
Document: {}
Score the document above. The answer should be 'Relevance score: X.' where X is a
number from 0-5.
0 means completely irrelevant, 5 means highly relevant and completely addresses
the query. Don't output anything else.
```

Figure 17: Prompt for reranking TheoQ.

---

**Prompt for reranking TheoT**

```
We want to find a document which uses the same mathematical process as this one:
{}
A document is relevant if it uses the same mathematical process as the query.
Document: {}
Score the document above. The answer should be 'Relevance score: X.' where X is a
number from 0-5.
0 means completely irrelevant, 5 means highly relevant and completely addresses
the query. Don't output anything else.
```

Figure 18: Prompt for reranking TheoT.

---

**Prompt for query decomposition**

```
You are a helpful assistant that generates multiple sub-questions related to an
input question.

The goal is to break down the input into a set of sub-problems / sub-questions
that can be answers in isolation.

Generate multiple search queries related to: {question}
Output (3 queries):
```

Figure 19: Prompt for reranking TheoT.

## B  Query Length Scaling

We adapt the query rewriting method in Su et al. (2024) to support controlled query length—specifically, we append a constraint of "Your answer must be written within {MAX_TOKENS} tokens." to the original query rewriting instruction and set the maximum number of output tokens correspondingly. The instruction can be found in Appendix A.4. We use GPT4O-MINI for query rewriting for the query scaling study.

## C  Query Decomposition

Query decomposition is another test-time technique has been shown to be effective on multi-hop retrieval tasks (Wang et al., 2024; Jin et al., 2025). We compare the retrieval performance for reasoning-intensive queries with query decomposition (Zhou et al., 2022) implemented by LangChain [3]. Query decomposition assumes the retrieval task can be decomposed into several simple sub-tasks. We show in Table 3 that this does not apply off-the-shelf to realistic reasoning-intensive retrieval tasks. In fact, query decomposition reduces performance from 0.121 to 0.105 with Nomic and 0.204 to 0.173 with GRIT-7B. This indicates that **an information-rich long query is better than several decomposed short queries on BRIGHT**.

## D  Details of Public Training Data

We include popular public training data , including MS MARCO (Nguyen et al., 2016), Natural Questions (Kwiatkowski et al., 2019), DUReader (He et al., 2017), FEVER (Thorne et al., 2018), HotpotQA (Yang et al., 2018), MIRACL (Zhang et al., 2023a), Mr. Tydi (Zhang et al., 2021), QUORA (DataCanary et al., 2017), Squad (Rajpurkar et al., 2016), T2Ranking (Xie

---

[3]https://www.langchain.com/

| Query Type | Num. Queries | BRIGHT nDCG@10 | |
|---|---|---|---|
| | | Nomic-v1.5 | GRIT-7B |
| Original Query | 1 | 0.121 | 0.204 |
| Query Decomposition | 3 | 0.105 | 0.173 |
| REASON-QUERY | 1 | 0.190 | 0.219 |

Table 3: Comparison between using an information-rich long query and 3 decomposed short queries. Both methods used GPT4 for query rewriting and decomposition.

et al., 2023), TriviaQA (Joshi et al., 2017). These datasets provide a diverse base data for adapting an autoregressive LLM into a bidirectional encoder for embedding tasks.

# E  Experimental Setup

In this section, we supplement with more details about the experimental setup.

## E.1  Contrastive Training Setup

**Training configuration.** We adapt Llama3.1-8B to use bi-directional attention mask and fine-tune the model using contrastive training with a batch size of 2048 for 1000 steps. We use a constant learning rate of 2e-5 with a warm-up ratio of 0.06. During training, we use both hard negatives and in-batch negatives. Following Muennighoff et al. (2024), we use a batch size of 2048 and apply techniques including GradCache (Gao et al., 2021) and cross-device negatives (Xiao et al., 2024b) (i.e., in-batch negatives gathered across devices) to enable a large global batch size for better training robustness.

## E.2  Reranker Setup

Prior work has demonstrated that traditional cross-encoder rerankers can degrade retrieval performance for reasoning-intensive queries, while reranking by using large language models (LLMs) generally improves performance (Su et al., 2024). Building on this observation, we investigated different reranking approaches and found that using the Qwen/Qwen2.5-32B-Instruct model in a zero-shot setting performs well (see Section G for a detailed comparison).

Our reranking method involves zero-shot prompting of the Qwen model to generate a score between 0 and 5 (the exact prompts are detailed in Section A.6). We then normalize this score to the range of 0 to 1 to obtain the reranker score $S_{reranker}$. To address potential score ties, we developed two scoring strategies:

1. QwenRerank+BM25: The final score is $\alpha S_{reranker} + S_{BM25}$, where $\alpha$ is a hyperparameter and $S_{BM25}$ is the BM25 score. To ensure that $S_{reranker}$ and $S_{BM25}$ are in the same range, we set $\alpha$ to 100 heuristically, without parameter tuning.

2. QwenRerank+retriever: The final score is $0.5 \times S_{reranker} + 0.5 \times S_{retriever}$, where $S_{retriever}$ is the normalized score from the base retrieval model.

## E.3  Evaluation Setup

**Evaluation & retrieval setup.** We use BRIGHT (Su et al., 2024) for evaluating reasoning-intensive information retrieval, employing nDCG@10 as the evaluation metric. For reasoning-intensive retrieval-augmented generation evaluation, we use MMLU (Hendrycks et al., 2020) and report the macro average across the 56 subjects. For MMLU and GPQA evaluation, we merge and deduplicate the top-1000 retrieved passages from MassiveDS-1.4T[4] (Shao et al., 2024) using Contriever (Izacard et al., 2021) to create an initial passage pool.

---

[4] https://huggingface.co/datasets/rulins/mmlu_searched_results_from_massiveds

| | Bio. | Earth. | StackExchange Econ. | Psy. | Rob. | Stack. | Sus. | Coding Leet. | Pony | Theorem-based AoPS | TheoQ. | TheoT. | Avg. |
|---|---|---|---|---|---|---|---|---|---|---|---|---|---|
| *Evaluate with original query* | | | | | | | | | | | | | |
| BM25 | 45.4 | 59.1 | 37.5 | 40.4 | 46.4 | 37.0 | 45.7 | 53.8 | 25.1 | 21.7 | 22.4 | 18.4 | 37.8 |
| Grit-7B | 62.9 | 56.9 | 51.7 | 59.3 | 41.1 | 57.5 | 55.6 | 65.9 | 37.5 | 26.4 | 44.8 | 60.2 | 51.6 |
| E5 | 57.6 | 55.3 | 43.3 | 57.8 | 38.6 | 46.9 | 49.5 | 65.0 | 27.1 | 22.8 | 44.4 | 63.1 | 47.6 |
| Qwen | 67.6 | 66.8 | 51.4 | 56.2 | 38.7 | 70.2 | 41.6 | 61.1 | 31.4 | 38.8 | 49.2 | 70.9 | 53.7 |
| REASONIR-8B | 70.5 | 65.5 | 58.4 | 69.4 | 47.8 | 69.6 | 64.5 | 69.4 | 31.1 | 37.4 | 52.2 | 63.9 | **58.3** |
| + Hybrid (BM25) | 67.2 | 70.6 | 59.5 | 60.1 | 52.2 | 65.5 | 58.9 | 63.3 | 42.0 | 32.5 | 38.4 | 46.5 | 54.7 |
| *Evaluate with GPT4 REASON-QUERY* | | | | | | | | | | | | | |
| BM25 | 85.2 | 77.1 | 50.4 | 67.1 | 50.3 | 58.8 | 61.5 | 38.8 | 45.2 | 15.4 | 38.8 | 49.2 | 53.2 |
| Grit-7B | 61.2 | 56.6 | 55.1 | 64.6 | 36.7 | 61.0 | 57.3 | 67.5 | 45.6 | 25.9 | 49.8 | 56.9 | 53.2 |
| E5 | 66.1 | 65.0 | 44.2 | 63.9 | 37.6 | 51.2 | 51.5 | 62.3 | 32.3 | 22.4 | 45.9 | 62.0 | 48.9 |
| Qwen | 76.6 | 72.7 | 56.7 | 71.9 | 42.5 | 58.2 | 65.2 | 65.3 | 32.3 | 22.4 | 48.6 | 72.8 | 57.1 |
| REASONIR-8B | 83.8 | 73.2 | 61.2 | 74.9 | 54.1 | 63.9 | 68.7 | 69.1 | 42.7 | 22.1 | 58.0 | 67.9 | 61.6 |
| + Hybrid (BM25) | 88.1 | 81.5 | 59.7 | 77.4 | 57.1 | 74.4 | 68.3 | 58.7 | 60.0 | 28.2 | 51.7 | 63.3 | **64.0** |
| *Evaluate with LLAMA3.1-8B-INSTRUCT REASON-QUERY* | | | | | | | | | | | | | |
| REASONIR-8B | 81.4 | 73.6 | 66.9 | 77.1 | 55.0 | 74.6 | 69.9 | 62.5 | 39.0 | 25.2 | 57.6 | 66.9 | 62.5 |

Table 4: Reasoning-intensive information retrieval performance on BRIGHT measured by Recall@100, which is strongly correlated with reranking performance. We highlight proprietary models with a gray background. "Hybrid (BM25)" refers to a hybrid version of REASONIR-8B, where we combine the similarity scores from REASONIR-8B and BM25 by interpolating them with a ratio of 0.5.

| | Bio. | Earth. | StackExchange Econ. | Psy. | Rob. | Stack. | Sus. | Coding Leet. | Pony | Theorem-based AoPS | TheoQ. | TheoT. | Avg. |
|---|---|---|---|---|---|---|---|---|---|---|---|---|---|
| *Evaluate with original query* | | | | | | | | | | | | | |
| BM25 | 50.2 | 67.9 | 46.0 | 47.8 | 53.4 | 50.5 | 52.7 | 57.5 | 67.2 | 28.3 | 22.6 | 19.9 | 47.0 |
| Grit-7B | 67.8 | 64.0 | 57.8 | 65.2 | 48.3 | 61.7 | 61.5 | 69.3 | 78.9 | 33.6 | 47.2 | 63.6 | 59.9 |
| E5 | 63.3 | 61.9 | 50.3 | 63.8 | 46.4 | 51.5 | 54.7 | 67.8 | 66.8 | 28.3 | 47.2 | 65.7 | 55.6 |
| Qwen | 72.6 | 73.3 | 58.6 | 63.4 | 46.1 | 73.9 | 47.0 | 64.8 | 71.0 | 46.4 | 51.8 | 73.5 | 61.9 |
| REASONIR-8B | 73.9 | 71.8 | 65.4 | 76.8 | 54.4 | 73.6 | 70.3 | 72.7 | 68.2 | 43.8 | 54.8 | 67.3 | 66.1 |
| + BM25 (Hybrid) | 72.5 | 77.8 | 67.9 | 68.6 | 59.7 | 69.5 | 65.8 | 66.9 | 84.3 | 40.1 | 40.3 | 49.0 | 63.5 |
| *Evaluate with GPT4 REASON-QUERY* | | | | | | | | | | | | | |
| BM25 | 88.3 | 83.2 | 59.8 | 75.1 | 57.8 | 62.5 | 68.9 | 41.9 | 87.4 | 20.8 | 41.5 | 51.7 | 61.6 |
| Grit-7B | 65.7 | 63.7 | 63.4 | 71.9 | 45.0 | 64.9 | 62.0 | 70.4 | 86.2 | 32.1 | 52.3 | 60.3 | 61.5 |
| E5 | 71.2 | 71.3 | 51.6 | 70.2 | 44.6 | 62.6 | 55.7 | 65.8 | 47.7 | 28.4 | 51.8 | 66.8 | 57.3 |
| Qwen | 79.9 | 79.5 | 65.0 | 79.2 | 51.0 | 72.0 | 70.9 | 69.3 | 74.1 | 29.2 | 54.2 | 76.9 | 66.8 |
| REASONIR-8B | 86.6 | 80.6 | 68.5 | 82.7 | 62.4 | 81.9 | 74.3 | 73.0 | 86.9 | 27.9 | 63.9 | 72.9 | **71.8** |
| + BM25 (Hybrid) | 89.9 | 87.3 | 67.9 | 85.1 | 65.5 | 77.7 | 74.6 | 62.1 | 95.5 | 35.1 | 54.6 | 66.0 | **71.8** |
| *Evaluate with LLAMA3.1-8B-INSTRUCT REASON-QUERY* | | | | | | | | | | | | | |
| REASONIR-8B | 84.1 | 80.6 | 73.3 | 84.1 | 62.7 | 79.3 | 74.8 | 66.3 | 75.6 | 30.9 | 60.9 | 70.3 | 70.2 |

Table 5: Reasoning-intensive information retrieval performance on BRIGHT measured by nDCG@10 of an oracle reranker that has access to the ground-truth query-document relevance scores. We highlight proprietary models with a gray background. "Hybrid (BM25)" refers to a hybrid version of REASONIR-8B, where we combine the similarity scores from REASONIR-8B and BM25 by interpolating them with a ratio of 0.5.

We then construct datastores using REASONIR-8B and baseline retrievers and compare their performance.

# F Additional Results on BRIGHT

In Table 4, we show additional BRIGHT results measured by Recall@100. Higher Recall@100 scores are more beneficial for post-retrieval reranking. We further computed the "Oracle nDCG@10" scores assuming a perfect reranker and report the performance in Table 5. The results show that REASONIR-8B significantly increases the theoretical upper bound for subsequent reranking, and indicate that there is still a large space for improvement for rerankers.

| | StackExchange | | | | | | | Coding | | Theorem-based | | | Avg. |
|---|---|---|---|---|---|---|---|---|---|---|---|---|---|
| | Bio. | Earth. | Econ. | Psy. | Rob. | Stack. | Sus. | Leet. | Pony | AoPS | TheoQ. | TheoT. | |
| QwenRerank (no tie-breaking) | 50.2 | 47.5 | 23.6 | 35.8 | 24.7 | 28.1 | 29.9 | 30.3 | 26.5 | 5.5 | 20.7 | 40.8 | 30.3 |
| QwenRerank+BM25 | 63.6 | 59.2 | 30.2 | 45.7 | 29.6 | 33.6 | 33.7 | 27.9 | 29.0 | 6.3 | 24.1 | 35.7 | 34.9 |
| QwenRerank+Retriever | 58.2 | 53.2 | 32.0 | 43.6 | 28.8 | 37.6 | 36.0 | 33.2 | 34.8 | 7.9 | 32.6 | 45.0 | 36.9 |

Table 6: Comparison of Different Tie-Breaking Methods

| | StackExchange | | | | | | | Coding | | Theorem-based | | | Avg. |
|---|---|---|---|---|---|---|---|---|---|---|---|---|---|
| | Bio. | Earth. | Econ. | Psy. | Rob. | Stack. | Sus. | Leet. | Pony | AoPS | TheoQ. | TheoT. | |
| Rank1-32b | 49.7 | 35.8 | 22.0 | 37.5 | 22.5 | 21.7 | 35.0 | 18.8 | 32.5 | 10.8 | 22.9 | 43.7 | 29.4 |
| QwenRerank+BM25 | 63.6 | 59.2 | 30.2 | 45.7 | 29.6 | 33.6 | 33.7 | 27.9 | 29.0 | 6.3 | 24.1 | 35.7 | 34.9 |

Table 7: Comparison of Rank1 and our proposed simplified Qwen32b Reranker

# G    Additional Reranker Results

In this section, we first compare our proposed LLM reranker with existing LLM reranker methods, showing our method is cheaper, training-free, and more performant when compared to existing LLM reranker baselines. Additionally, we ablate the effect of using REASONIR-8B as the base retriever when compared with GRIT-7B and BM25. We demonstrate that REASONIR-8B results in better performance when combined with a reranker.

## G.1    Breaking Ties in QwenRerank

As explained in Section E.2, we break ties in the reranker scores using two potential strategies. In Table 6 we show results from using different tie-breaking strategies. As seen from the table, the resutls are worse when the reranker scores are used directly without tie-breaking.

## G.2    Rank1 vs QwenRerank

Weller et al. (2025) train a reranking model for reasoning-intensive queries by obtaining reasoning traces from DeepSeek-R1 (Guo et al., 2025) on MS-Marco queries and documents, and using these traces to train a distilled Qwen2.5 reranker. Their model, Rank1, first generates reasoning traces and then outputs relevance scores for given query-document pairs. In contrast, our proposed QwenRerank method directly outputs a score without generating any explicit reasoning tokens during inference and is much faster. As shown in Table 7, our approach outperforms the Rank1 method and we therefore use QwenRerank as our default reranker.

## G.3    Comparison of Candidates from Different Retrieval Systems

In this section, we evaluate candidate sets generated by different retrieval methods to assess their potential for downstream reranking performance. Table 8 shows that ReasonIR retrieves a better candidate pool for the Qwen-32b reranker.

## G.4    Reranker Generalization

In this section, we compare the performance of rerankers in combination with different underlying retrievers. Specifically, the rerankers always rerank the top-100 scored documents from retrievers. From these experiments, we observe an interesting phenomenon related to reranker generalization.

In the previous section, REASONIR-8B provides the highest Recall@100 and Oracle results in Tables 4 and 5, respectively. This indicates that REASONIR-8B generally provides the most useful documents among all retrievers. However, when REASONIR-8B serves as the basis for Rank1-7B (providing its top-100 retrieved documents as reranking candidates), the performance is worse than both Rank1-7B (BM25) and REASONIR-8B by itself. Similarly, the reranker results with Grit-7B are also worse than those with BM25, despite Grit-7B outperforming BM25 when used as a standalone retriever. This indicates that certain rerankers may

| | Bio. | Earth. | StackExchange | | | | | Coding | | Theorem-based | | | Avg. |
|---|---|---|---|---|---|---|---|---|---|---|---|---|---|
| | | | Econ. | Psy. | Rob. | Stack. | Sus. | Leet. | Pony | AoPS | TheoQ. | TheoT. | |
| | | | | *Rerank the top-100 candidates with QwenRerank+BM25* | | | | | | | | | |
| BM25 | 63.6 | 59.2 | 30.2 | 45.7 | 29.6 | 33.6 | 33.7 | 27.9 | 29.0 | 6.3 | 24.1 | 35.7 | 34.9 |
| GritLM | 53.7 | 56.9 | 32.2 | 44.6 | 29.5 | 32.7 | 32.8 | 35.0 | 29.8 | 9.9 | 29.6 | 43.0 | 35.8 |
| ReasonIR-8b | 57.7 | 59.5 | 30.7 | 47.3 | 29.6 | 36.2 | 34.3 | 32.7 | 37.3 | 7.6 | 28.0 | 41.0 | 36.8 |
| | | | | *Rerank the top-100 candidates with QwenRerank+Retriever* | | | | | | | | | |
| BM25 | 53.8 | 54.2 | 24.4 | 38.7 | 19.0 | 27.8 | 26.3 | 19.3 | 17.8 | 4.0 | 19.3 | 20.9 | 27.1 |
| GritLM | 51.2 | 51.9 | 31.5 | 40.3 | 26.6 | 33.6 | 33.8 | 34.6 | 28.5 | 8.8 | 30.4 | 44.6 | 34.6 |
| ReasonIR-8b | 58.2 | 53.2 | 32.0 | 43.6 | 28.8 | 37.6 | 36.0 | 33.2 | 34.8 | 7.9 | 32.6 | 45.0 | 36.9 |

Table 8: Comparison of Candidates from different Retrieval Systems

| | Bio. | Earth. | StackExchange | | | | | Coding | | Theorem-based | | | Avg. |
|---|---|---|---|---|---|---|---|---|---|---|---|---|---|
| | | | Econ. | Psy. | Rob. | Stack. | Sus. | Leet. | Pony | AoPS | TheoQ. | TheoT. | |
| | | | | *Evaluate with* GPT4 REASON-QUERY | | | | | | | | | |
| BM25 | 53.6 | **53.6** | 24.3 | 38.6 | 18.8 | 22.7 | 25.9 | 19.3 | 17.7 | 3.9 | 20.2 | 18.9 | 26.5 |
| **REASONIR-8B** | 43.6 | 42.9 | **32.7** | 38.8 | 20.9 | 25.8 | 27.5 | 31.5 | 19.6 | 7.4 | 33.1 | 35.7 | **29.9** |
| Rank1-7B (BM25) | 48.8 | 36.7 | 20.8 | 35.0 | 22.0 | 18.7 | **36.2** | 12.7 | 31.2 | 6.3 | 23.7 | 37.8 | 27.5 |
| Rank1-7B (Grit-7B) | 34.7 | 28.4 | 20.9 | 32.7 | 15.6 | 19.5 | 21.8 | 11.9 | 34.0 | 7.1 | 28.8 | 35.0 | 24.2 |
| Rank1-7B (REASONIR-8B) | 42.8 | 33.3 | 23.0 | 33.8 | 16.7 | 20.0 | 28.1 | 9.3 | 32.7 | 6.7 | **36.0** | 31.2 | 26.1 |
| **QWENRERANK (REASONIR-8B)** | **58.2** | 53.2 | 32.0 | **43.6** | **28.8** | **37.6** | 36.0 | **33.2** | **34.8** | 7.9 | 32.6 | **45.0** | **36.9** |

Table 9: Comparing the performance of Rerankers with different retrievers. We focus on Rank1-7B using top-100 candidates provided by BM25, Grit-7B, and REASONIR-8B. Although Grit-7B and REASONIR-8B outperform or perform comparably as BM25 when used as a standalone retriever, existing rerankers such as Rank1-7B may not be able to fully leverage their benefits, as Rank1-7B performs the best with candidates from BM25. Note that GPT4 REASON-QUERY is provided for retrievers to generate the top-100 candidates, but this information is not available to the rerankers.

not be able to fully leverage the benefits of stronger retrievers, such as REASONIR-8B. Upon further investigation, as shown in Section G.5, there exists a distribution shift between the documents retrieved by BM25 and REASONIR-8B. As Rank1's training is based on learning to rank hard negatives from Tevatron[5] (which utilizes mostly BM25 and CoCondenser), we hypothesize that **the distribution shifts in the top-100 candidates from different retrievers might cause performance degradation for the rerankers that are trained for some specific retrievers**.

### G.5 Retrieval Overlap between BM25 and REASONIR-8B

Although REASONIR-8B outperforms BM25 by a large margin on both normal queries and rewritten queries (with GPT4 reasoning). As shown in Table 10, across 12 subtasks in BRIGHT, BM25 and REASONIR-8B only share about 28.2% overlap in the top-100 retrieved documents. In terms of the gold documents, the overlap ratio is only up to 53.5%. Therefore, they provide quite dissimilar sets of top-100 documents, complementing each other. This might be one of the reasons that the hybrid model (REASONIR-8B + BM25) works better on BRIGHT. In addition, the set difference in terms of top-100 documents will also cause a change in the basis for the downstream rerankers.

## H  Additional Downstream Results

We present the results on MMLU and GPQA in Table 11.

---
[5] https://huggingface.co/datasets/Tevatron/msmarco-passage-aug

| | | | StackExchange | | | | | Coding | | Theorem-based | | | Avg. |
|---|---|---|---|---|---|---|---|---|---|---|---|---|---|
| | Bio. | Earth. | Econ. | Psy. | Rob. | Stack. | Sus. | Leet. | Pony | AoPS | TheoQ. | TheoT. | |
| Top-100 | 42.5 | 36.6 | 36.4 | 35.4 | 25.0 | 27.1 | 35.6 | 16.2 | 25.3 | 12.5 | 20.2 | 25.2 | 28.2 |
| Gold | 81.9 | 75.1 | 53.0 | 66.7 | 48.3 | 57.2 | 66.3 | 42.6 | 49.7 | 11.0 | 50.0 | 40.3 | 53.5 |

Table 10: Ratio of overlapping documents retrieved from BM25 and REASONIR-8B. "Top-100" means the average percentage of overlapping documents in the top-100 retrieved documents. "Gold" means the percentage of overlapping gold (ground-truth) documents out of the top-100 retrieved documents from each retriever that are also gold. The scores are expressed as percentages (%).

| Retriever | Query Type | MMLU | GPQA |
|---|---|---|---|
| Closed-book | - | 71.1 | 31.3 |
| Contriever | Original question | 72.0 | 36.4 |
| GRIT-7B | Original question | 74.1 | 32.3 |
| Search Engine | Original question | - | 33.8 |
| **REASONIR-8B** | Original question | 75.0 | **38.4** |
| Contriever | REASON-QUERY | 72.8 | 31.3 |
| GRIT-7B | REASON-QUERY | 74.7 | 30.8 |
| Search Engine | REASON-QUERY | - | 36.4 |
| **REASONIR-8B** | REASON-QUERY | **75.6** | 35.4 |

Table 11: Performance on NLP tasks where the knowledge sources are unknown or unavailable.

# I  Augmenting Training on Reasoning Rewritten Queries

## I.1  Training with Reasoning Rewritten Queries (RQ)

We are curious if directly training the retriever on rewritten queries can help it better utilize query rewriting technique at test time. Therefore, we also experiment with a new setting where we supplement training data with reasoning rewritten queries (REASON-QUERY). Specifically, we generate REASON-QUERY queries using the reasoning-intensive synthetic data as RQ. Due to computational constraints, we reuse the positives and negatives of the original queries for reasoning queries, assuming that query rewriting does not alter the query's relevance to the documents. We leave it to future work to explore better methods for curating positive and negative documents for reasoning queries.

We first compare the model performance when training on each *individual* type of synthetic data. We find that **every type of synthetic data can enhance retrieval performance** when used in conjunction with the public training set. Specifically, VL data achieves the highest score of 23.2 when compared with HQ and RQ when evaluated with the original queries. Meanwhile, RQ achieves the highest performance of 28.5 on reasoning queries when compared with the other two types of data. Adding HQ in addition to the public training data only improves the performance on the original queries from 19.6 to 20.9, while it does not enhance performance on reasoning queries. Additionally, HQ under-performs VL on both types of evaluation queries. We hypothesize that this is because HQ data is both longer and harder than the public training data, making it too challenging for the model to directly learn from it. In contrast, the difficulty level of VL data is more moderate in helping the model adapt to length without being overly challenging.

In conclusion, **without combining different types of synthetic data, training on RQ is more beneficial for improving REASON-QUERY evaluation performance, while training on VL is more helpful for enhancing the original query evaluation performance**.

| | | | StackExchange | | | | | Coding | | Theorem-based | | | Avg. |
|---|---|---|---|---|---|---|---|---|---|---|---|---|---|
| | Bio. | Earth. | Econ. | Psy. | Rob. | Stack. | Sus. | Leet. | Pony | AoPS | TheoQ. | TheoT. | |
| *w/o CoT query rewriting* | | | | | | | | | | | | | |
| LLAMA3.1-8B | 12.5 | 6.5 | 7.7 | 7.7 | 3.9 | 7.5 | 8.6 | 22.0 | 17.1 | 10.5 | 7.4 | 2.0 | 9.5 |
| Public | 21.4 | 30.3 | 17.8 | 24.7 | 18.6 | 18.8 | 18.8 | 30.0 | 6.7 | 12.1 | 21.4 | 15.2 | 19.6 |
| Public+HQ | 21.0 | 31.3 | 18.4 | 25.1 | 15.7 | 18.4 | 14.3 | 34.1 | 5.2 | 9.5 | 33.7 | 24.4 | 20.9 |
| Public+RQ | 20.0 | 26.8 | 19.7 | 26.9 | 17.4 | 19.9 | 15.6 | 31.4 | 9.3 | 13.0 | 28.7 | 21.2 | 20.8 |
| Public+VL | 28.4 | 35.8 | 22.5 | 28.4 | 18.4 | 19.5 | 18.7 | 34.5 | 12.3 | 11.4 | 24.4 | 23.6 | 23.2 |
| Public+EQVL | 26.8 | 33.8 | 23.4 | 30.1 | 21.1 | 21.9 | 21.5 | 31.0 | 6.5 | 10.1 | 20.9 | 20.2 | 22.3 |
| Public+HQVL | 26.2 | 31.4 | 23.3 | 30.0 | 18.0 | 23.9 | 20.5 | 35.0 | 10.5 | 14.7 | 31.9 | 27.2 | **24.4** |
| Public+RQVL | 21.3 | 31.1 | 26.5 | 29.5 | 18.8 | 21.7 | 19.9 | 28.1 | 9.8 | 15.9 | 25.0 | 24.0 | 22.6 |
| Public+HQRQVL | 26.6 | 30.7 | 18.9 | 27.8 | 19.2 | 23.1 | 19.2 | 36.8 | 9.2 | 10.3 | 33.0 | 28.1 | 23.6 |
| *w/ CoT query rewriting* | | | | | | | | | | | | | |
| LLAMA3.1-8B | 41.3 | 25.1 | 16.8 | 17.3 | 8.7 | 10.7 | 15.7 | 6.8 | 32.3 | 0.9 | 12.3 | 4.0 | 16.0 |
| Public | 40.3 | 42.1 | 26.0 | 37.7 | 20.8 | 22.6 | 22.7 | 32.3 | 13.5 | 7.0 | 29.5 | 30.4 | 27.1 |
| Public+HQ | 37.4 | 42.7 | 26.8 | 35.3 | 18.2 | 22.1 | 20.0 | 35.0 | 14.7 | 6.7 | 34.1 | 32.7 | 27.1 |
| Public+RQ | 38.1 | 41.2 | 29.1 | 37.3 | 21.4 | 25.8 | 24.8 | 31.6 | 15.4 | 8.1 | 33.2 | 35.3 | 28.5 |
| Public+VL | 33.8 | 41.3 | 28.9 | 40.2 | 20.6 | 24.2 | 25.9 | 34.7 | 19.6 | 4.8 | 32.5 | 29.2 | 28.0 |
| Public+EQVL | 37.9 | 42.1 | 30.6 | 40.0 | 22.1 | 25.6 | 27.4 | 31.8 | 15.5 | 6.1 | 27.3 | 28.7 | 27.9 |
| Public+HQVL | 43.6 | 42.9 | 32.7 | 38.8 | 20.9 | 25.8 | 27.5 | 31.5 | 19.6 | 7.4 | 33.1 | 35.7 | **29.9** |
| Public+RQVL | 30.1 | 40.6 | 37.1 | 38.0 | 22.0 | 27.1 | 25.7 | 33.7 | 20.6 | 8.6 | 34.1 | 34.7 | 29.4 |
| Public+HQRQVL | 35.7 | 40.0 | 27.2 | 35.9 | 19.4 | 24.7 | 22.3 | 34.3 | 13.5 | 8.8 | 35.2 | 36.0 | 27.8 |

Table 12: Ablation on the training data. We show the nDCG@10 BRIGHT performance with models trained on different data sources. Each model is trained for 1000 steps. LLAMA3.1-8B is the base auto-regressive LLM we use as the init checkpoint. When used as an embedding model, we use a bi-directional attention mask on the inputs and apply average pooling on the hidden states of the last layer to obtain the embeddings. "Public" stands for public training data collected from existing training datasets. HQ, RQ, VL, and EQ refer to hard queries, reasoning queries, varied-length distillation data, and easy queries, respectively.

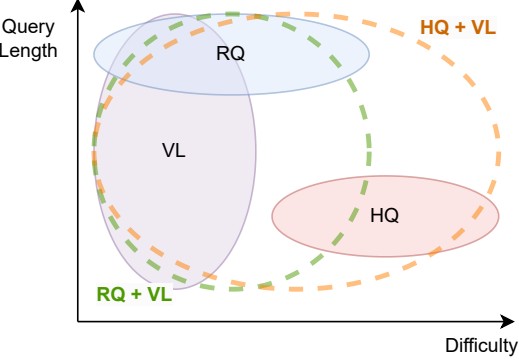

Figure 20: An illustration of our hypothesis on the effect of different data mixture strategies.

**The combination of HQ and VL yields the best performance on both original query and REASON-QUERY evaluations.** We then combine different types of synthetic data and study their performance. We find that combining HQ and VL yields the best performance, with an nDCG@10 score of 24.4 on original queries and 29.9 on reasoning queries. Intriguingly, we found that combining HQ and VL performs even better than combining RQ and VL on REASON-QUERY evaluation. This indicates that the benefits brought by length generalization and reasoning capability to handle complex information in the queries are sufficient to enhance the performance on REASON-QUERY. We visualize our hypothesis on the effect of data mixture in Figure 20, showing that the coverage of query difficulty and length distribution impacts the final performance. However, when training on HQ separately, there was not enough data for length generalization for REASON-QUERY, which led to inferior performance. We also experiment with a mix of all three types of synthetic data, as shown in Table 2, where we subsample the HQ data generated by half of the document pool and subsample the RQ data generated by the remaining half. This is to avoid overlapping positive and negative documents. We found that the model trained on this set of data performs

worse than when trained on $\overline{\text{HQ}}$ and $\overline{\text{VL}}$ only, indicating that it is more important to include $\overline{\text{HQ}}$ than $\overline{\text{RQ}}$ when mixing with $\overline{\text{VL}}$ data.

## J  Test-time Compute Comparison between Retriever and Reranker

In this section, we compare the test-time compute required for bi-encoder retriever (e.g., REASONIR-8B) and cross-encoder reranker (e.g., Rank1 (Weller et al., 2025)) from the perspective of both time complexity and FLOPS. Suppose that we are given a query $q$ with $L_q$ tokens at test-time. We assume the base LLM is a transformer coupled with KV cache during inference. For FLOPS, we use the approximation from Kaplan et al. (2020), where encoding/decoding each token costs about $2N$ and $N$ is the number of non-embedding parameters of the model. We omit the FLOPS required for performing similarity search over the indexed datastore, as this procedure for retrievers is also applied for rerankers (because they are based on the retriever output). The results are shown in Tables 13 and 15, where REASONIR-8B can be more than $100\times$ faster than Rank1. We describe our analysis as follows.

| Model name | Time complexity | FLOPS |
|---|---|---|
| Retriever | $O(L_q^2)$ | $2NL_q$ |
| Reranker | $O\left(k((L_q + L_d)^2 + (L_q + L_d)L_o + L_o^2)\right)$ | $2Nk(L_q + L_d + L_o)$ |

Table 13: Test-time time complexity and FLOPS comparison between bi-encoder retriever and cross-encoder reranker.

**Bi-encoder retriever.** Since the datastore has already been pre-indexed, the time complexity for encoding the query at test time is $O(L_q^2)$ for a fixed-sized model. Since the query length is $L_q$, the retriever approximately incurs a total of $2NL_q$ FLOPS.

**Cross-encoder reranker.** Let $k$ be the number of documents to be reranked at test time after obtaining them from the retriever. Let $L_q$ and $L_d$ denote the length of the query and the document, respectively. To encode $k$ query-document pairs for reranking, the basic time complexity is $O\left((k(L_q + L_d)^2\right)$. In addition, the reranker could reason by generating additional tokens before outputting the similarity score (Weller et al., 2025). Let $L_o$ denote the average output length. The time complexity of a transformer-based reranker with KV-cache becomes $O\left(k((L_q + L_d)^2 + (L_q + L_d)L_o + L_o^2)\right)$. The compute cost in terms of FLOPS is $2Nk(L_q + L_d + L_o)$ for processing all related tokens, which is significantly more expensive than the cost of a retriever model $2NL_q$, especially when $k$ is large (typically $k = 100$) or $L_o$ is long (e.g., Rank1-7B outputs about 300 tokens before generating the similarity score).

**FLOPS Calculation.** To calculate the FLOPS, we simply plug the corresponding values to the formula. In particular, REASONIR-8B has about 7.0 billion non-embedding parameters; Rank1-7B has about 6.5 billion non-embedding parameters; and Rank1-32B has 31 billion. For different lengths, we use the estimated statistics shown in 14. We estimate the FLOPS in Table 15. In practice, Rank1-7B can be a few hundred times slower than REASONIR-8B with normal reasoning-intensive queries. Moreover, REASONIR-8B (with GPT4 REASON-QUERY) is more than $200\times$ faster than Rank1-32B (with normal query) and outperforms it on BRIGHT.

## K  Additional Related Works

**Test-time scaling for RAG.** Optimal test-time scaling (Snell et al., 2024), which significantly increases the quality and efficiency of LLMs, requires balancing the compute across various components. Prior work has explored enhancing RAG at test time from the perspective of (1) *Datastore scaling* by increasing its size and diversity (Shao et al., 2024); (2) *Context-length*

| Normal query len. | GPT-4 REASON-QUERY len. | Doc len. | Avg. # of output tokens |
|---|---|---|---|
| 128 | 1024 | 300 | 300 |

Table 14: Variable values for FLOPS Computation.

| Model name | Test-time FLOPS |
|---|---|
| REASONIR-8B (normal query) | $\alpha = 1.9 \times 10^{12}$ |
| REASONIR-8B (GPT4 REASON-QUERY) | $\beta = 15.4 \times 10^{12} = 5.6\alpha$ |
| Rank1-7B (normal query) | $946 \times 10^{12} = 61.6\beta$ |
| Rank1-32B (normal query) | $4.5 \times 10^{15} = 293.9\beta$ |

Table 15: Test-time FLOPS comparison between REASONIR-8B and Rank1 models for **biology** subtask of BRIGHT. We denote the cost of REASONIR-8B (GPT4 REASON-QUERY) as a $\beta$ unit. The estimated statistics for the query, document, and reranker output lengths are documented in Table 14.

*scaling* by increasing the number of retrieved documents in the context (Jiang et al., 2024; Xu et al., 2023; Yue et al., 2024); (3) *Reranker scaling* by increasing the number of top candidate documents to rerank and increasing the number of thinking tokens to rerank documents after retrieval (Weller et al., 2025); (4) *Iteration scaling* by increasing the number of rounds to retrieve and respond (Trivedi et al., 2022; Yue et al., 2024). Our work studies the test-time scaling for RAG from a 5th perspective—*rewriter scaling*—and highlights the benefit of allocating more test-time compute for rewriting queries (Liu & Mozafari, 2024; Ma et al., 2023) with reasoning, which critically affects the retrieval quality (Su et al., 2024).

## L   Qualitative Examples

In this section, we show the advantages of REASONIR-SYNTHESIZER by qualitatively comparing the hard queries HQ generated by our approach vs. easy queries EQ generated by a simple baseline.

### L.1   Qualitative Examples for VL and HQ in Detail

In this section, we provide the qualitative examples in full for examples used in Figure 20. The example for VL is shown in Table 16 and the example for HQ is shown in Table 17.

### L.2   Qualitative Examples for Traditional Retrieval and Reasoning-intensive Retrieval

We illustrate the difference between simple retrieval vs. reasoning-intensive retrieval with a qualitative example in Figure 21.

### L.3   Qualitative Examples for Hard Query HQ Generation for Different Subjects

Hard queries (HQ) are an important part of our synthetic data. We show examples of hard queries that we generate for different domains in REASONIR-SYNTHESIZER in tables 18, 19, 20, 21, 22 and 23.

| | |
|---|---|
| Query | A former employee, Sarah, is suing her ex-employer, a large corporation, for wrongful termination. Sarah claims she was fired after reporting a supervisor's unethical behavior to human resources. The corporation argued that the termination was due to restructuring, but Sarah argues this was a pretext for retaliation. The case hinges on determining whether the corporation's stated reason for termination was the true reason or if retaliation was involved and if so, whether that constitutes wrongful termination under the law. The corporation's lawyer argues that the decision to terminate Sarah's employment was not based on her report but instead due to the ongoing restructuring efforts within the company. However, evidence shows that the restructuring efforts were limited to Sarah's department alone, raising questions about the legitimacy of the stated reason for termination. Can the court hold the corporation liable for wrongful termination in this scenario? |
| Positive Doc | In the landmark case of Crawford v. Metropolitan Government of Nashville and Davidson County (2009), the Supreme Court of the United States addressed the issue of retaliation after reporting workplace misconduct. Vicky Crawford, an employee of the Metropolitan Government of Nashville, reported that a supervisor had sexually harassed her. Shortly after the report, Crawford was terminated. The court held that Title VII's retaliation provision prohibits 'any action that well might have dissuaded a reasonable worker from making or supporting a charge of discrimination.' The court sided with Crawford, ruling that the termination was a result of retaliation for reporting the misconduct. The court also established that employers cannot terminate an employee in response to a protected activity, such as reporting workplace harassment, without facing potential liability for retaliation. Crawford v. Metropolitan Government set a precedent in identifying retaliation claims, making it a pivotal ruling in similar wrongful termination cases that involve allegations of retaliation after reporting misconduct in the workplace. |
| Hard Negative | The issue of corporate restructuring has become a recurring theme in the modern business world. As companies continue to adapt to the ever-changing market, restructuring has become a common strategy to stay competitive. However, the manner in which companies restructure can have a significant impact on employees. On one hand, restructuring can lead to the elimination of redundant positions, resulting in a more streamlined and efficient workforce. On the other hand, restructuring can be used as a euphemism for layoffs or downsizing, leading to job insecurity and instability for employees. In recent years, there has been an increased focus on the ethics of corporate restructuring, particularly with regards to the treatment of employees during this process. While some argue that restructuring is necessary for a company's survival, others contend that it can be used as a means to exploit employees and undermine their rights.In the wake of the 2008 financial crisis, corporate restructuring became a common practice as companies sought to cut costs and remain afloat. However, this led to widespread job losses and economic instability for many individuals and families. In response, governments and regulatory bodies have implemented measures to protect employees' rights during restructuring, such as requiring companies to provide adequate notice and severance packages.Despite these efforts, the issue of corporate restructuring remains a contentious one, with ongoing debates about its impact on employees and the broader economy. While it is clear that restructuring can be a necessary evil in some cases, it is equally important to ensure that companies prioritize their employees' well-being and rights during this process.Regulatory bodies have also established guidelines to ensure that companies are transparent and fair in their restructuring efforts. For instance, companies must often provide adequate notice to employees, as well as offer support and training to those who are being let go. By prioritizing employees' needs and rights, companies can minimize the negative impacts of restructuring and create a more positive outcome for all stakeholders involved. |

Table 16: Example $\overline{VL}$ query generation using REASONIR-SYNTHESIZER. The query is long, containing 167 tokens. The positive document is about the legal case that is directly relevant to the query. The hard negative describes the "corporate restructuring", a keyword used in the query, but is not directly relevant to answer the query.

| Query | A researcher is studying the sleep patterns of a group of individuals who work night shifts. The researcher notices that these individuals tend to have difficulty falling asleep during the day and experience fatigue during their work hours. What could be the primary factor contributing to this phenomenon, and how might it be related to the body's natural oscillations? |
|---|---|
| Positive Doc | A circadian rhythm (/səˈkeɪdiən/), or circadian cycle, is a natural oscillation that repeats roughly every 24 hours. Circadian rhythms can refer to any process that originates within an organism (i.e., endogenous) and responds to the environment (is entrained by the environment). Circadian rhythms are regulated by a circadian clock whose primary function is to rhythmically co-ordinate biological processes so they occur at the correct time to maximise the fitness of an individual. Circadian rhythms have been widely observed in animals, plants, fungi and cyanobacteria and there is evidence that they evolved independently in each of these kingdoms of life. The term circadian comes from the Latin circa, meaning "around", and dies, meaning "day". Processes with 24-hour cycles are more generally called diurnal rhythms; diurnal rhythms should not be called circadian rhythms unless they can be confirmed as endogenous, and not environmental. Although circadian rhythms are endogenous, they are adjusted to the local environment by external cues called zeitgebers (from German Zeitgeber (German: [ˈtsaɪtˌɡeːbɐ]; lit.'time giver')), which include light, temperature and redox cycles. In clinical settings, an abnormal circadian rhythm in humans is known as a circadian rhythm sleep disorder. |
| Hard Negative | During the night shift, individuals often experience an increase in body temperature, which can lead to discomfort and difficulty maintaining focus. This increase in temperature is usually highest in the late evening, around 10-11 pm, and gradually decreases as the night progresses. Body temperature is known to be controlled by the hypothalamus, which acts as the body's thermostat. The hypothalamus responds to changes in the body's core temperature to cool the body or warm it up through various mechanisms. These processes are an essential element of the body's natural responses, but understanding their relationship to the observed phenomenon of difficulty during the night shift is somewhat complex. Factors affecting the body's core temperature include the ambient temperature, intensity of workouts, and personal characteristics. The hypothalamus responds to the ambient temperature and helps maintain the body's core temperature. When the ambient temperature is high, sweating and other heat-loss mechanisms are activated, whereas low ambient temperatures result in the body conserving heat through the constriction of blood vessels near the skin. A higher intensity of workouts or engaging in activities that utilize more muscle mass increases body temperature.The thermoregulation response also varies across individuals based on characteristics including age, sex, and fitness level. However, the night shift affects body temperature, the same way regardless. It does not exhibit the same variations between individuals as natural temperature regulation might.Various studies have demonstrated the adverse effects of elevated body temperatures on sleep and vigilance. Revealing a correlation, rather than a causality between high body temperatures during night shifts and the subjective experience of discomfort with day-time sleep. These various factors interacting shows some overlaps with but also confusion with the issues experienced by people during the night shift study prompt. |

Table 17: Example HQ query generation using REASONIR-SYNTHESIZER. The query about sleeping patterns for a group of individuals with night shifts is relatively long, containing 66 tokens. The positive document is about the circadian rhythm, offering a scientific explanation to the problem. On the other hand, the hard negative, despite containing keywords such as "night shifts" and attempting to explain the scenario with "body temperature", is not correct as it is not the primary factor contributing to the phenomenon. In addition, it is irrelevant to "natural oscillations" asked by the query.

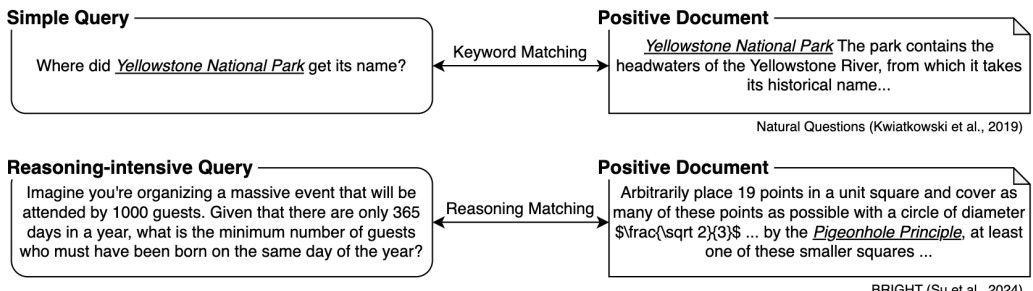

Figure 21: Relevant documents for simple queries can be retrieved easily via keyword matching, while reasoning is required to retrieve for reasoning-intensive queries.

| EQ | What is the function of cone cells in the human eye and how do they contribute to color vision? |
|---|---|
| HQ | A researcher is studying the visual perception of a person with a rare genetic condition that affects the structure of their cone cells. The person's retina has an unusually high concentration of cones sensitive to medium-wavelength light, but a lower concentration of cones sensitive to short-wavelength and long-wavelength light. How might this affect the person's ability to perceive colors in their environment, and what potential advantages or disadvantages might this have in different visual tasks? |
| Doc | Cone cells or cones are photoreceptor cells in the retinas of vertebrates' eyes. They respond differently to light of different wavelengths, and the combination of their responses is responsible for color vision. Cones function best in relatively bright light, called the photopic region, as opposed to rod cells, which work better in dim light, or the scotopic region. Cone cells are densely packed in the fovea centralis, a 0.3Å mm diameter rod-free area with very thin, densely packed cones which quickly reduce in number towards the periphery of the retina. Conversely, they are absent from the optic disc, contributing to the blind spot. There are about six to seven million cones in a human eye (vs 92 million rods), with the highest concentration being towards the macula. Cones are less sensitive to light than the rod cells in the retina (which support vision at low light levels), but allow the perception of color. They are also able to perceive finer detail and more rapid changes in images because their response times to stimuli are faster than those of rods. Cones are normally one of three types: S-cones, M-cones and L-cones. Each type expresses a different opsin: OPN1SW, OPN1MW, and OPN1LW, respectively. These cones are sensitive to visible wavelengths of light that correspond to short-wavelength, medium-wavelength and longer-wavelength light respectively. Because humans usually have three kinds of cones with different photopsins, which have different response curves and thus respond to variation in color in different ways, humans have trichromatic vision. Being color blind can change this, and there have been some verified reports of people with four types of cones, giving them tetrachromatic vision. The three pigments responsible for detecting light have been shown to vary in their exact chemical composition due to genetic mutation; different individuals will have cones with different color sensitivity. |

Table 18: Example query generation using REASONIR-SYNTHESIZER for a document of cone cells in the field of **biology**. While EQ directly asks for the function of cone cells, HQ frames a complex problem about how the concentration of cone cells affects human color perception (e.g., color blindness).

| EQ | What is the primary purpose of shivering in warm-blooded animals? |
|---|---|
| HQ | A patient is experiencing a high fever, but instead of feeling hot, they report feeling extremely cold. What physiological mechanism could be causing this discrepancy between the patient's actual body temperature and their perceived temperature? |
| Doc | Shivering (also called shuddering) is a bodily function in response to cold and extreme fear in warm-blooded animals. When the core body temperature drops, the shivering reflex is triggered to maintain homeostasis. Skeletal muscles begin to shake in small movements, creating warmth by expending energy. Shivering can also be a response to fever, as a person may feel cold. During fever, the hypothalamic set point for temperature is raised. The increased set point causes the body temperature to rise (pyrexia), but also makes the patient feel cold until the new set point is reached. Severe chills with violent shivering are called rigors. Rigors occur because the patient's body is shivering in a physiological attempt to increase body temperature to the new set point. |

Table 19: Example query generation using REASONIR-SYNTHESIZER for a document of shivering in the field of **biology**. While EQ directly asks for the function of shivering, HQ frames an analytic problem about using hypothalamic set point, a concept associated with shivering, to explain why people often feel cold when they are experiencing fever.

| EQ | Write a function to find the first letter that appears twice in a given string of lowercase English letters. |
|---|---|
| HQ | A researcher is analyzing a dataset of strings, each representing a sequence of lowercase English letters. The researcher wants to identify the first letter that appears at least twice in each string. However, the dataset is corrupted, and some strings may contain non-alphabetic characters or uppercase letters. How can the researcher modify their algorithm to handle these corrupted strings while still identifying the first letter to appear twice in the valid strings? |
| Doc | (see code below) |

```
def first_letter_to_appear_twice(s: str) -> str:
 """Given a string `s` consisting of lowercase English letters, return _the
 first letter to appear **twice**_.

**Note**:

* A letter `a` appears twice before another letter `b` if the **second**
occurrence of `a` is before the **second** occurrence of `b`.
* `s` will contain at least one letter that appears twice.

**Example 1:**

**Input:** s = "abccbaacz "
**Output:** "c "
**Explanation:**
The letter 'a' appears on the indexes 0, 5 and 6.
The letter 'b' appears on the indexes 1 and 4.
The letter 'c' appears on the indexes 2, 3 and 7.
The letter 'z' appears on the index 8.
The letter 'c' is the first letter to appear twice, because out of all the
letters the index of its second occurrence is the smallest.

**Example 2:**

**Input:** s = "abcdd "
**Output:** "d "
**Explanation:**
The only letter that appears twice is 'd' so we return 'd'.

**Constraints:**

* `2 <= s.length <= 100`
* `s` consists of lowercase English letters.
* `s` has at least one repeated letter."""

 occurrences = [0] * 26
 for c in s:
 occurrences[ord(c) - ord('a')] += 1
 if occurrences[ord(c) - ord('a')] == 2:
 return c
 return '?'
```

Table 20: Example query generation using REASONIR-SYNTHESIZER for **Leetcode**. EQ *repeat the question from the docstring of the coding example.* In contrast, HQ complicates the question by adding string corruption to the problem setting.

| | |
|---|---|
| EQ | How many distinct integers will be present on the board after 10^9 days if we start with a number n? |
| HQ | A mathematician is studying the properties of a sequence of numbers generated by a specific rule. The rule states that for each number x in the sequence, all numbers i such that $1 \le i \le x$ and x % i == 1 are added to the sequence. If the sequence starts with a single number n, what is the maximum number of distinct numbers that can be present in the sequence after a large number of iterations, assuming n is a positive integer less than or equal to 100? |
| Doc | ```
def distinct_numbers(n):
  """You are given a positive integer `n`, that is initially placed on a
  board. Every day, for `109` days, you perform the following procedure:

  * For each number `x` present on the board, find all numbers `1 <= i <= n`
  such that `x % i == 1`.
  * Then, place those numbers on the board.

  Return _the number of **distinct** integers present on the board after_
  `109` _days have elapsed_.

  **Note:**

  * Once a number is placed on the board, it will remain on it until the end.
  * `%` stands for the modulo operation. For example, `14 % 3` is `2`.

  **Example 1:**

  **Input:** n = 5
  **Output:** 4
  **Explanation:** Initially, 5 is present on the board.
  The next day, 2 and 4 will be added since 5 % 2 == 1 and 5 % 4 == 1.
  After that day, 3 will be added to the board because 4 % 3 == 1.
  At the end of a billion days, the distinct numbers on the board will be 2,
  3, 4, and 5.

  **Example 2:**

  **Input:** n = 3
  **Output:** 2
  **Explanation:**
  Since 3 % 2 == 1, 2 will be added to the board.
  After a billion days, the only two distinct numbers on the board are 2 and
  3.

  **Constraints:**

  * `1 <= n <= 100`"""

  return n - 1
``` |

Table 21: Example query generation using REASONIR-SYNTHESIZER for **Leetcode**. EQ generates an invalid question without clear specifications about the objective. In contrast, HQ produces a valid question with a complete scenario and increases the difficulty by asking for "a large number of iterations" instead of for "109" iterations.

| EQ | What is the formula for a production function? |
|---|---|
| HQ | A manufacturing company is considering increasing its labor force to boost production. The company's current production level is 1000 units per day with 10 workers. If the company adds 2 more workers, the production level increases to 1200 units per day. However, when the company adds 2 more workers again, the production level only increases to 1300 units per day. What can be inferred about the company's production function, and what might happen if the company continues to add more workers? |
| Doc | `Production function is a concept that explains the relationship between`
`quantities of input and the output. Its formula is Q=f(K,L). Production`
`function is of two types: short-run and long-run, depending on the number of`
`fixed factors. Products can be of three kinds: Total product, Average`
`product,`
`and marginal product. There is a specific relationship between total product`
`and marginal product as well as marginal product and average product.`
`Production function is also used in the Law of Variable Proportions which`
`states that as the quantity of one variable input increases, the total`
`product`
`first increases at an increasing rate, then decreases at a decreasing rate,`
`and ultimately falls and the marginal product becomes negative.` |

Table 22: Example query generation using REASONIR-SYNTHESIZER in the field of **economics**. EQ generates a simple question for recalling the formula. In contrast, HQ generates a reasoning-intensive question that requires using the knowledge of production function to analyze a realistic scenario.

| EQ | What is the relationship between inflation and unemployment according to the Phillips Curve? |
|---|---|
| HQ | A country is experiencing a period of rapid economic growth, with GDP increasing by 5% annually. However, the inflation rate has also risen to 4%, causing concerns among policymakers. The central bank is considering implementing contractionary monetary policies to reduce inflation, but is worried about the potential impact on employment. Using the relationship between inflation and unemployment, what is the likely outcome of reducing inflation through contractionary monetary policies, and what tradeoff might the policymakers face? |
| Doc | `Key Takeaways`
`The Phillips Curve is a graph that shows the tradeoff between inflation and`
`unemployment.`
`Under the Phillips Curve, high inflation is accompanied with low`
`unemployment, and low inflation is accompanied by high unemployment.`
`Policymakers use the Phillips Curve to manage the tradeoff between`
`inflation and unemployment.`
`Some economists think that the Phillips Curve doesn't reflect monetary`
`factors and implies that economic growth is always inflationary.`
`How Does the Phillips Curve Work?`
`The Phillips Curve is a graph that plots unemployment against inflation. In`
`general, it shows that inflation and unemployment have an inverse`
`relationship. When inflation is high, unemployment tends to be low, and`
`when inflation is low, unemployment tends to be high.` |

Table 23: Example query generation using REASONIR-SYNTHESIZER in the field of **economics**. EQ generates a simple question with direct reference to the keyword "Philips Curve". In contrast, HQ asks about the impact of reducing inflation on unemployment, which requires analysis to be linked to the Philips Curve.

