# OpenReview forum: "ReasonIR: Training Retrievers for Reasoning Tasks"
_colmweb.org/COLM/2025/Conference — COLM 2025_

### Official Review · Reviewer_EF6c · 2025-05-06

**Rating:** 8
**Confidence:** 4
**Ethics Flag:** 1

**Summary:**

The paper presents ReasonIR, a specialized retrieval model for reasoning-intensive tasks. ReasonIR leverages synthetic training data comprising varied-length queries and hard-negatives generated via LLM prompting. These synthetic data are experimentally shown to be challenging for vanilla retrievers, which ReasonIR addresses. By training on this dataset, ReasonIR achieves state-of-the-art performance on the BRIGHT benchmark while maintaining greater efficiency than alternative LLM rerankers.

**Questions To Authors:**

Additional questions:

Q1) Could the authors provide details on the training data size of competing methods, e.g., GritLM vs. ReasonIR? What is the benefit when training GritLM with the same data as ReasonIR does?

Q2) Would ReasonIR have negative impact on simpler retrieval tasks? For example, what is the performance of ReasonIR(+VL+HQ) on the original MS-MARCO/Natural Questions/HotpotQA data?

**Reasons To Accept:**

Strengths:

S1) ReasonIR excels at reasoning-intensive tasks, where retrieval is challenging due to query complexity, by relying on carefully generated synthetic data. The motivation for using synthetic data is well-grounded in the pilot study presented in Section 3.

S2) ReasonIR is effective under different test-time compute scenarios: it achieves the best performance among baselines both when using the original query and when incorporating query rewriting techniques on the BRIGHT dataset. ReasonIR also outperforms search engines in MMLU and GPQA benchmarks.

S3) Ablations on the proposed synthetic data generation techniques validate the importance of varied-length queries and hard-negatives in tasks such as coding and theorem-based ones (Table 2).

**Reasons To Reject:**

Weakness:

The paper lacks an analysis of training scaling laws based on the proposed synthetic data generation techniques. While lines 232-233 state that "training data include 1,383,877 public training samples, 244,970 VL samples, and 100,521 HQ samples" the authors do not investigate how optimal this specific ratio (public/VL/HQ) is. This raises questions about whether increasing the proportion of VL or HQ samples would lead to improved downstream performance.

For example, Table 2 demonstrates that the generated data are not equally beneficial across all tasks—specifically, in the stack-exchange subset, HQ samples provide minimal to no improvement over the Public training samples alone. Given that performance increases with more samples (as shown in Public+VL+EQ), it might be more effective to directly increase the VL set to approximately 350,000 total samples rather than incorporating HQ or EQ samples.

Such studies are important to better understand the cases where ReasonIR's synthetic data mix can provide the best improvements.

---

> ### Author Response · Authors · 2025-06-03
>
> We thank the reviewer for the insightful and comprehensive comments. We also thank the reviewer for recognizing our motivation, the efficiency and effectiveness in retrieval and RAG, and the insights from the ablation study. We address the comments below.
>
> **Weaknesses**:
>
> **Q.** The paper lacks an analysis of training scaling laws and data combination ratios based on the proposed synthetic data generation techniques.
>
> **A.** We agree with the reviewer that studying the scaling behavior of the size and the composition of datasets is an important research direction. Within our computational budget, we prioritized ablations on data composition in Table 2 of the main paper. However, the scaling study is computationally intensive. For example, each training run requires 640 GPU hours. Data generation at the current scale requires thousands of GPU hours. Therefore, a full scaling study is out of our budget. Thus, we briefly discussed scaling studies for future work in Section 7. Nonetheless, we provide a preliminary study to check the potential of scaling below for this rebuttal.
>
> Specifically, we subsample existing data into 50% of the full ReasonIR training data and retrain the model on this subsampled data. As shown in the table below, reducing the data to half drastically reduces the final performance. Based on the results, we hypothesize that further scaling up the data may lead to additional performance improvements. While our work initiates efforts to generate synthetic data for reasoning-intensive retrieval tasks, we leave it to future work to study the corresponding scaling laws and optimize the ratio of the data due to computational constraints.
>
> **Table. BRIGHT performance. ReasonIR-8B is trained on our full dataset (public+VL+HQ), and ReasonIR-8B-0.5scale is trained on public data and 50% randomly sampled VL and HQ data (public+50%VL+50%HQ).**
>
> | | Bio. | Earth. | Econ. | Psy. | Rob. | Stack. | Sus. | Leet. | Pony | AoPS | TheoQ. | TheoT. | Avg. |
> |---|---|---|---|---|---|---|---|---|---|---|---|---|---|
> | Evaluate with original query|
> | ReasonIR-8B | 26.2 | 31.4 | 23.3 | 30.0 | 18.0 | 23.9 | 20.5 | 35.0 | 10.5 | 14.7 | 31.9 | 27.2 | 24.4 |
> | ReasonIR-8B-0.5scale | 19.1 | 25.6 | 19.9 | 26.4 | 18.6 | 24.0 | 17.7 | 33.7 | 6.1 | 12.0 | 25.7 | 20.7 | 20.8 |
> | Evaluate with GPT4 Reason-query|
> | ReasonIR-8B | 43.6 | 42.9 | 32.7 | 38.8 | 20.9 | 25.8 | 27.5 | 31.5 | 19.6 | 7.4 | 33.1 | 35.7 | 29.9 |
> | ReasonIR-8B-0.5scale | 35.5 | 39.1 | 29.0 | 37.0 | 21.4 | 32.0 | 23.9 | 28.4 | 23.7 | 7.1 | 30.3 | 33.0 | 28.4 |
>
> **Questions**:
>
> **Q1.** Could the authors provide details on the training data size of competing methods, e.g., GritLM vs. ReasonIR? What is the benefit when training GritLM with the same data as ReasonIR does?
>
> **A1.** GritLM differs from ReasonIR in multiple aspects, such as the size and openness of the data, as well as the training objective. Overall, GRIT uses 2,216,784 samples that are not fully open-sourced while we use 1,729,368 samples that are fully open-sourced.
>
> Firstly, the data used by GritLM is not fully open-sourced. As shown in the table below, GritLM partially uses GPT4-generated proprietary synthetic data, which is closed-sourced. On the other hand, all data used by ReasonIR is fully open-sourced.
> Secondly, GritLM also performs generative modeling using the Tulu V2 dataset, while ReasonIR focuses on training only the bi-encoder retriever. The GritLM paper claims that this additional training object helps GRIT achieve better performance on embedding/retrieval tasks.
>
> The details of the training data composition between GritLM and ReasonIR are shown in the two tables below.
>
> **Table. Data Composition for GritLM and ReasonIR.**
> |Subset|Quantity|Open-sourced|
> |---|---|---|
> |**GritLM**|
> |Public|1,383,877|Yes|
> |Tulu-V2|326,154|Yes|
> |GPT4-synthetic|506,753|No|
> |Total|2,216,784|-|
> |**ReasonIR**|
> |Public|1,383,877|Yes|
> |VL|244,970|Yes|
> |HQ|100,521|Yes|
> |Total|1,729,368|-|
>
> As stated above, the key differences between our work and GRIT are the data and the objective. Since we focus on embedding training, we did not apply the language modeling objective. We follow GRIT's training setup and include their publicly available training data in our training set. As shown in Table 2, training on this public data alone results in 19.6 nDCG@10 on BRIGHT. Adding VL and HQ data increases performance to 24.4. Therefore, we believe that training GritLM with ReasonIR data should improve its reasoning-intensive retrieval performance if we had access to the full training data (including the proprietary ones). We leave studying the best combination of these training techniques to future work.

---

> > ### Author Response · Authors · 2025-06-03
> >
> > **Q2.** Would ReasonIR have negative impact on simpler retrieval tasks? For example, what is the performance of ReasonIR(+VL+HQ) on the original MS-MARCO/Natural Questions/HotpotQA data?
> >
> > **A2.** For the rebuttal, we investigate the effect of our data generation pipeline on non-reasoning retrieval tasks by comparing the nDCG@10 score of ReasonIR-8B trained with all three types of data with the model trained with just the public data. As shown in the Table below, adding VL+HQ data splits significantly improves the performance of the retriever trained only with public data and BM25, indicating that training on VL+HQ data is generally helpful on non-reasoning retrieval tasks.
> >
> > **Table. Evaluation of ReasonIR on non-reasoning retrieval tasks.**
> >
> > ||MSMARCO|HotPotQA|NQ|
> > |-|-|-|-|
> > |BM25|21.90|32.08|28.49|
> > |ReasonIR (Public only)|13.68|56.39|15.49|
> > |ReasonIR (Public+VL+HQ)|32.67|62.96|52.97|

---

> ### Comment · Reviewer_EF6c · 2025-06-03
>
> Thank you for your response. My questions have been addressed and **I have increased my score**. Please include these discussions in the final paper.

---

> > ### Author Response · Authors · 2025-06-09
> >
> > We thank the reviewer for acknowledging the helpfulness of our rebuttal and for increasing the score! We will incorporate these discussions in the final paper.

---

### Official Review · Reviewer_MLfg · 2025-05-11

**Rating:** 6
**Confidence:** 4
**Ethics Flag:** 1

**Summary:**

This paper proposes a LLM-based bi-encoder model - ReasonIR-8B for reasoning intensive queries. Compared with existing reasoning reranker (Rank1), it enjoys both efficiency and accuracy improvements due to the advantange of dense retrieval over cross-encoder. The author proposes a novel dataset creation schema consisted of varied length document,query pair generation as well as hard negative generation. In experiments, the performance of ReasonIR-8B stands out on both (1) original query and (2) GPT-rewritten queries.

**Questions To Authors:**

1. In Table 1, why the Rank1 method does not work with the original query?
2. Is the gap between Rank1 smaller on non-reasoning retrieval benchmarks? In other words, does author think the reranker is not neccessary for reasoning queries?
3. In Section "Reasoning-worthy seed document selection", the author mentioned the corpus is selected from BRIGHT benchmark paper. Is there a knowledge leakage to the evaluation benchmark?
4. I noticed the performance degration of GPT4 Reason Query on Coding and Theorem on the proposed method? Is this due to the generated training query is closer to the test query?

**Reasons To Accept:**

1. This paper spearhead the LLM-based representation learning and reasoning-intensive retrieval tasks right on the trending "test-time compute" and reasoning LLMs.
2. The performance is superior against multiple recent LLM-based retriever/retranker.

**Reasons To Reject:**

1. The performance of the paper seems to rely on in-domain "high quality synthetic data", the out-of-distribution/generation capability is not examined.

---

> ### Author Response · Authors · 2025-06-03
>
> We thank the reviewer for providing insightful comments and for recognizing both the novelty of our work and the efficiency and effectiveness of our model for reasoning-intensive retrieval. Our responses are as follows:
>
> **Weaknesses**
>
> **Q1.** The performance of the paper seems to rely on in-domain "high quality synthetic data", the out-of-distribution/generation capability is not examined.
>
> **A1.** We thank the reviewer for initiating this discussion. In Section 5.2, we evaluated our retriever's generalization ability on RAG tasks (MMLU and GPQA) where no documents exist for synthetic data generation. As shown in Figure 1, our retriever brings consistent gains on both reasoning-intensive RAG tasks (4.5% gains on MMLU and 7.2% gains on GPQA compared with the closed-book baseline), outperforming GRIT-7B and you.com search engine baselines.
>
> For BRIGHT, we agree that the domains of documents used to curate HQ data align with the benchmark. The 12 domains covered by seed documents are general and diverse, including math, biology, earth science, and economics. Importantly, we did not use any BRIGHT test queries to curate synthetic training data. The generation pipeline only accessed documents created by browsing and crawling STEM websites. We are optimistic that the benefits of our data can be generalized to other STEM IR benchmarks.
>
> Additionally, though not explored here, our work could potentially be extended as a domain-adaptation method for resource-limited domains, as noted by Reviewer 4Dkf. For example, when accessing private documents, one could curate private training data following our approach without human-annotated queries. We will edit our paper to include this discussion on our method's generalization and add our discussion on its potential application as an unsupervised domain adaptation method in the discussion section.
>
> **Questions**
>
> **Q1.** In Table 1, why the Rank1 method does not work with the original query?
>
> **A1.** Rank1 can also work with the original query. We apologize for not including it originally because the performance was not reported in its paper. In response to this question, we evaluate rank1’s performance (using top-100 candidates from BM25) on the original query in the table below. ReasonIR, having an average nDCG@10 score of 24.4, outperforms rank1’s 21.0 by a larger margin in this scenario (compared to the case with GPT4 Reason-query, where the nDCG@10 score is 27.5 for rank1 and 29.9 for ReasonIR). We will include the additional results in the revision.
>
> **Table. Reranking performance of Rank1 on BRIGHT with original queries.**
>
> ||Bio.|Earth.|Econ.|Psy.|Rob.|Stack.|Sus.|Leet.|Pony|AoPS|TheoQ.|TheoT.|Avg.|
> |-|-|-|-|-|-|-|-|-|-|-|-|-|-|
> |ReasonIR|26.2|31.4|23.3|30|18|23.9|20.5|35|10.5|14.7|31.9|27.2|24.4|
> |rank1|31.5|34.4|16.3|25.3|21.1|17.1|25.2|17.5|23.4|9.8|16.4|14.4|21.0|
>
> **Q2.** Is the gap between Rank1 smaller on non-reasoning retrieval benchmarks? In other words, does author think the reranker is not neccessary for reasoning queries?
>
> **A2.** We answer the two questions separately:
>
> To answer the first question, we compare ReasonIR with rank1 on the subset of BEIR (a non-reasoning retrieval benchmark) tasks reported by rank1. As shown in the table below, ReasonIR-8B outperforms all rank1 variants. We will include these results in our next version of the paper.
>
> **Table. Comparing ReasonIR with Rank1 on a subset of BEIR tasks. Rank1 uses top-100 candidates retrieved via BM25.**
>
> |Model|ArguA|ClimF|DBP|FiQA|NFCorp|SciDoc|SciFact|Touche|TrecC|Avg.|
> |-|-|-|-|-|-|-|-|-|-|-|
> |Rank1-7B|42.8|15.0|38.9|39.5|36.2|17.2|77.2|22.8|81.9|40.9|
> |Rank1-14B|45.3|16.2|37.4|37.9|35.8|17.9|77.0|27.1|78.2|41.0|
> |Rank1-32B|57.6|15.8|40.7|41.8|36.9|19.6|76.8|19.9|81.9|41.7|
> |ReasonIR-8B|56.4|31.8|39.4|48.8|37.5|22.7|75.2|20.1|75.9|**45.3**|
>
> For the second question, we think it is important to study and improve rerankers for reasoning-intensive retrieval, as it can provide complementary gains to the base bi-encoder retriever. As shown in Section 4.6, our simple ReasonIR-Rerank boosts ReasonIR’s performance w/ GPT4 reasoning from 29.9 to 36.9. We advocate future work to develop more efficient and performant rerankers for reasoning-intensive retrievers to result in further gains on top of ReasonIR.

---

> > ### Author Response · Authors · 2025-06-03
> >
> > **Q3.** In Section "Reasoning-worthy seed document selection", the author mentioned the corpus is selected from BRIGHT benchmark paper. Is there a knowledge leakage to the evaluation benchmark?
> >
> > **A3.** We acknowledge the reviewer’s concern on potential data leakage and would love to discuss how it is unlikely in our case. BRIGHT has two components: the corpus of 1M documents, and the test set of (query, gold document) pairs which is 4k out of those 1M. The documents in the BRIGHT benchmark are crawled by the BRIGHT authors from diverse STEM websites. We assume it is reasonable to have access to this large-scale STEM corpus, as long as we do not use any test query information for data generation.
> > We fully understand the concern from the reviewer that having the gold documents in the pool may introduce indirect information leakage about the distribution of the gold documents. We show that such information does not necessarily lead to better performance—in Table 2, we show that generating easy queries (EQ) using the same set of documents does not bring any further gains on top of VL data. In fact, including EQ in the training set harms the performance, though it contains the same amount of information about the documents as our HQ data.
> >
> > **Q4.** I noticed the performance degration of GPT4 Reason Query on Coding and Theorem on the proposed method? Is this due to the generated training query is closer to the test query?
> >
> > **A4.** We would appreciate the reviewer's clarification to help us understand their question—ReasonIR's performance improves on GPT4 Reason-query for Coding and Theorem-based tasks on average (from 22.8 to 25.6 and from 24.6 to 25.4, respectively). While performance degrades on two subtasks (LeetCode and AoPS), ReasonIR's gains in Pony, TheoQ, and TheoT remain significant. This performance drop from GPT4 rewriting isn't specific to ReasonIR—BM25 and Contriever also have drops on LeetCode, and BM25 and GritLM show similar drops on AoPS. We hypothesize that the performance drop results from GPT4's inability to generate helpful search queries for these two tasks.
> >
> > To understand this better, we qualitatively examine the documents, original query, and GPT4 Reason-query on LeetCode. We find that the documents in LeetCode’s datastore often contain only coding problem definitions and solutions without analysis (e.g., what algorithm the code implements). Therefore, even though GPT4 effectively expands the query by finding relevant algorithms to solve the problem, it is still challenging to match the expanded query with solution-only documents that do not explicitly contain the name of the algorithms.
> >
> > For example, ReasonIR successfully retrieves for the original query “778. Swim in Rising Water”, which shares a similar problem definition with the gold document “1631. Path With Minimum Effort”. However, after rewriting the query with GPT4, the retrieval fails. Specifically, GPT4 identifies that the query problem can be solved using algorithms such as “binary search on time” and “depth-first search (DFS)”. However, it can be challenging for existing dense models to directly understand that the gold document relies on algorithms such as “binary search” and “DFS”, since it has not been analyzed. Therefore, the valuable information provided by GPT4 cannot be fully leveraged. As a result, it can confuse the retriever to retrieve semantically similar but irrelevant instances like “1091. Shortest Path in Binary Matrix”. We will add the details of this example with the above discussion in our next paper version.

---

> ### Author Response · Authors · 2025-06-09
>
> We appreciate the reviewer's thoughtful feedback and the time invested in reviewing our submission. As the discussion period draws to a close, we remain available to provide any additional clarification or address further questions that would be helpful!

---

### Official Review · Reviewer_P7d1 · 2025-05-12

**Rating:** 7
**Confidence:** 3
**Ethics Flag:** 1

**Summary:**

This paper presents an approach to train retrievers for reasoning tasks. The main contribution of this work is introducing a data generation pipeline. REASONIR is trained using data generated by this pipeline, and achieves sate-of-the-art on BRIGHT and RAG tasks. The paper is well written, but has some presentation issues, while some experimental details are missing.

**Questions To Authors:**

1. In certain cases, generating long queries can make the task easier because of high lexical match with the relevant document. How do you control that? Is the LLM able to always avoid high lexical match? Would other, smaller LLMs be able to deal with this complexity?

**Reasons To Accept:**

* Good motivation experiment and insights in Section 3.
* Interesting idea for hard negative generation in Section 4.4.
* Strong results, efficient method.
* Thorough experimental results in the appendix

**Reasons To Reject:**

Presentation clarity:
* Section 1: An example would help to illustrate what reasoning is in this context.
* Section 4.2 is not self-contained: it is fine referencing the appendix but the description in the main text should be sufficient to understand the main idea.  Same for the “reasoning intensive document-to-query generation” paragraph in Section 4.3.
* Section 5.3. EQ appears in Table 2 before it gets properly defined.


Experiments
* Lacks definition of baselines in Table 1; OpenAI, Voyage, Google are not defined, and the rest of the baselines are only defined by citation without providing training details and ensuring fair comparison with the proposed approach
* An analysis on why the combination with the sparse retriever is beneficial would be interesting to show cases where REASONIR fails and can be improved

---

> ### Author Response · Authors · 2025-06-03
>
> We thank the reviewer for providing insightful and detailed suggestions. We have updated the draft to fix the reviewer’s suggestions on presentation clarity, including
> * Adding an example reasoning-intensive query from BRIGHT to the introduction to help clarify the reasoning in the context of retrieval.
> * Providing a high-level summary of the content in the appendix when referring to it for details.
> * Moving the definition of EQ before Table 2.
> The change will appear in the next version of our submission.
>
> We address the reviewer’s other questions below:
>
> **Q1.** Lacks definition of baselines in Table 1; OpenAI, Voyage, Google are not defined, and the rest of the baselines are only defined by citation without providing training details and ensuring fair comparison with the proposed approach.
>
> **A1.** We apologize for the missing baseline definitions and thank the reviewer for pointing this out. The baselines and numbers originate from the official BRIGHT benchmark. We will supplement closed-source model references and note the score sources of these baselines in our next revision.
>
> **Q2.** An analysis on why the combination with the sparse retriever is beneficial would be interesting to show cases where REASONIR fails and can be improved.
>
> **A2.** We agree that a qualitative analysis will help to further improve the paper, since our quantitative results (Appendix G.3) show that BM25 and ReasonIR are retrieving different sets of documents, with only a 50% overlap.
> To answer the question, we analyzed failure cases of ReasonIR on BRIGHT. For certain queries, critical keywords appear in both the original query and the GPT4-reason query explicitly, allowing simple BM25 keyword matching to successfully retrieve relevant documents. However, as ReasonIR encodes the full semantic meaning of queries/documents as a dense retrieval model, the target keyword's semantic meaning can get diluted by other content, causing potential failure. As a result, we hypothesize that ReasonIR may not always effectively encode rare or specialized terms for long-form inputs.
>
> For example, in a BRIGHT psychology task, a query about "kinetic synesthesia" (a perceptual phenomenon in which people may experience colors when listening to music) is easily addressed by BM25 via keyword matching. However, ReasonIR focuses broadly on "perception"-related documents rather than "synesthesia". A sparse retriever like BM25 can effectively increase the influence of certain rare keywords that ReasonIR may fail to encode. Alternatively, improving representations of rare but important concepts could further enhance ReasonIR's performance. We will include this discussion in our next revised version.
>
> **Q3.** In certain cases, generating long queries can make the task easier because of high lexical match with the relevant document. How do you control that? Is the LLM able to always avoid high lexical match? Would other, smaller LLMs be able to deal with this complexity?
>
> **A3.** We would appreciate more clarification–is the question asking about the difficulty control for our HQ or VL data?
>
> For VL data, the focus of including it is to extend the retriever’s context length. Despite potential high lexical overlap, increased length makes VL data more challenging than public data—as shown in Figure 3, both GRIT-7B and BM25 have higher error rates on VL data than on public data.
> If the reviewer is asking about Reason-query: we use LLM to rewrite queries to make searching for targeted documents easier. Since query rewriting aims to make retrieval easier, higher lexical overlap is desirable in this process.
>
> For HQ data, our data-generation prompt includes "avoiding high lexical overlap with relevant documents when generating hard queries" as a requirement (Appendix A.1). The quantitative assessment of the HQ data is discussed in Section 4.5, where we show that the HQ data generated using our prompt has the highest BM25 error rate of 53%, indicating the lowest lexical overlap.
> To compare small LLMs' effectiveness at generating HQ data and lexical match levels, we generated hard queries for 1200 seed documents using Llama3.1-8B with the same prompt and compared their difficulty levels (described in Section 4.5) with Llama3.1-70B queries. As shown below, the queries generated from the 8B model have significantly lower difficulty. In particular, the low BM25 error indicates high lexical match between generated queries and positive documents, meaning the smaller LLM fails to follow our data-generation instructions precisely.
>
> **Table. Comparing the difficulty of queries generated by small and large LLMs.**
> |Model|Llama3.1-8B|Llama3.1-70B|
> |-|-|-|
> |BM25 Error|21.0|53.0|
> |GRIT-7B Error|7.5|42.3|

---

> > ### Comment · Reviewer_P7d1 · 2025-06-06
> >
> > Thank you for the clarifications. On A1 and A2, looking forward to the discussion in the revised version. On A3, my question was about the VL data. I am keeping my original score.

---

> > > ### Author Response · Authors · 2025-06-09
> > >
> > > We thank the reviewer for their clarification. For VL data, we agree that longer queries tend to have more lexical overlap. As shown in Figure 3, VL data has a lower BM25 error rate (37.3%) compared to our HQ data (53.0%). However, having long queries with possibly high lexical overlap is not necessarily problematic for several reasons:
> > >
> > > **Enabling the retriever to be also good at handling long queries with lexical overlap matters:**
> > > There are several practical cases where the queries could be lengthy with lexical overlap. For example,
> > > 1. **Natural long queries**: When legitimate long queries have lexical matches, retrievers should still exploit these matches effectively.
> > > 2. **Query rewriting preparation**: Our VL data prepares the retriever for query rewriting techniques (introduced in our Section 2) that can possibly increase query length and lexical overlap to improve document retrieval.
> > >
> > > **Longer queries may remain challenging despite lexical overlap:**
> > > Although longer queries create more opportunities for lexical matching, they are not necessarily easier to handle because they may introduce more distractors—there could be more documents that have lexical overlap with the long queries. This pattern holds for both natural/rewritten queries and our VL data, demonstrating that VL data accurately reflects real-world query challenges. Figure 3 in our submission shows that both GRIT-7B and BM25 have higher error rates on VL data (11.3% and 37.3%, respectively) compared to existing public training data (3.9% and 25.1%, respectively).
> > >
> > > **Empirical validation:**
> > > Our results in Table 2 show that training on VL data only also improves model performance, confirming that handling long queries with potential lexical overlap is a valuable capability for modern retrieval systems.
> > >
> > > Our previous response showed that small models struggle with the complex instructions required for HQ data generation. However, VL data generation uses much simpler instructions—we don't require reasoning-intensive queries or low lexical overlap. Therefore, small models could potentially generate VL data at lower cost. However, we hypothesize that larger models may still be preferable for generating diverse data requiring knowledge that small models lack. We leave exploring small model data generation to future work.
> > >
> > > We appreciate the reviewer's thoughtful feedback, and we remain available to provide any additional clarification or discussion that would be helpful!

---

### Official Review · Reviewer_4Dkf · 2025-05-13

**Rating:** 7
**Confidence:** 4
**Ethics Flag:** 1

**Summary:**

This work proposes to create synthetic dataset to train a retriever model for retrieval augmented generation (RAG) in large language models specifically designed for reasoning tasks. The accuracies in retrieval is crucial in RAG but the limitation in the diversity in the retrieval data and hard negatives is the major bottleneck in training a model. This work leverages large language models for synthetic data creation by generating queries for seed documents and hard negative documents for each query. Experiments show gains in retrieval accuracies measured on a standard benchmark dataset, BRIGHT, and also present gains on end-task of MMLU and GPQA.

**Questions To Authors:**

See the reasons to reject.

**Reasons To Accept:**

- This work directly addresses the limitation of training data by creating a synthetic dataset specifically designed for retrieval considering two aspects, length variation and intensiveness in reasoning. The approach sounds reasonable, and the proposed approach clearly show gains when compared with other baselines.
- This work might have an impact for future studies in RAG especially on the domain with limited data.

**Reasons To Reject:**

- It is not clear the impact of the quality of the synthetic data, since there exist no studies to verify it manually. If possible, better to run an experiment by trading the quality of the synthetic data and the actual retrieval accuracies by, e.g., injecting noises into the synthetic data  by an LLM.
- It is also interesting to measure the impact of the synthetic dataset sizes to see if more data will be helpful or not.

---

> ### Author Response · Authors · 2025-06-03
>
> We thank the reviewer for acknowledging our contribution on data generation and impact on future RAG research on data-limited domains. We also appreciate the reviewer’s insightful questions with regard to data quality and scale. We address the comments below.
>
> **Q1.** It is not clear the impact of the quality of the synthetic data, since there exist no studies to verify it manually. If possible, better to run an experiment by trading the quality of the synthetic data and the actual retrieval accuracies by, e.g., injecting noises into the synthetic data by an LLM.
>
> **A1.** Following the reviewer's suggestion, we manually inject noise into the data. Specifically, we ask an LLM to corrupt 10% of the HQ queries using the prompt below:
>
> ```bash
> System:
> You are a helpful assistant.
> Your task is to corrupt the queries by rewriting them in one or more of the following ways:
> 1. Inject small factual errors or misleading words before or after the main query.
> 2. Add some off-topic or unrelated information to the query.
> 3. Introduce multiple grammatical errors or typos in the query.
>
> After rewriting the query, output the new query strictly following the json format below.
> ```json
> {
>     "corrupted_query": <your rewritten query here>.
> }
>
> User:
> Here is the query you need to corrupt:
>
> {query}
>
> ```
>
> The other 90% HQ queries remain unchanged.
>
> We train a new model, ReasonIR-8B-perturbed, on this new data using the same training configuration as ReasonIR-8B. As shown in the Table below, injecting noise into 10% of the HQ data slightly harms the averaged performance—the BRIGHT performance slightly decreased from 24.4 to 24.0. We hypothesize that it is because our query perturbation prompt, which adds minor errors and irrelevant information, only slightly harms the data quality. We will add these new results and discussions into the next version of our paper.
>
> **Table. BRIGHT performance. ReasonIR-8B-perturbed is trained on the same dataset as ReasonIR-8B, but with 10% of the data perturbed.**
>
> | | Bio. | Earth. | Econ. | Psy. | Rob. | Stack. | Sus. | Leet. | Pony | AoPS | TheoQ. | TheoT. | Avg. |
> |---|---|---|---|---|---|---|---|---|---|---|---|---|---|
> | BM25 | 19.2 | 27.1 | 14.9 | 12.5 | 13.5 | 16.5 | 15.2 | 24.4 | 7.9 | 6.0 | 13.0 | 6.9 | 14.8 |
> | Contriever | 9.2 | 13.6 | 10.5 | 12.1 | 9.5 | 9.6 | 8.9 | 24.5 | 14.7 | 7.2 | 10.4 | 3.2 | 11.1 |
> | GritLM-7B | 25.0 | 32.8 | 19.0 | 19.9 | 17.3 | 11.6 | 18.0 | 29.8 | 22.0 | 8.8 | 25.1 | 21.1 | 20.9 |
> | ReasonIR-8B | 26.2 | 31.4 | 23.3 | 30.0 | 18.0 | 23.9 | 20.5 | 35.0 | 10.5 | 14.7 | 31.9 | 27.2 | 24.4 |
> | ReasonIR-perturbed | 27.1 | 34.1 | 26.7 | 29.7 | 19.1 | 23.7 | 19.6 | 31.8 | 11.2 | 12.8 | 26.7 | 25.8 | 24.0 |
>
>
> **Q2.** It is also interesting to measure the impact of the synthetic dataset sizes to see if more data will be helpful or not.
>
> **A2.** We agree that studying synthetic data scaling trends would provide useful insights for future work. We provide a preliminary scaling study by subsampling 50% of the full ReasonIR training data and retraining the model (see ReasonIR-8B-0.5scale in the table). As shown below, halving the data reduces nDCG@10 performance from 24.4 to 20.8 with original queries and from 29.9 to 28.4 with GPT4 Reason-query. We conjecture that generating more training data could yield further improvements.
>
> However, the scaling study is computationally intensive. For example, each training run requires 640 GPU hours. Data generation at the current scale requires thousands of GPU hours. Therefore, a full scaling study is out of our budget. Within our computational budget, we prioritized ablations on data composition in Table 2. We briefly discussed scaling studies for future work in Section 7 and will expand within our budget. We will add these new results to the next version of the paper.
>
> **Table. BRIGHT performance. ReasonIR-8B is trained on the full dataset (public+VL+HQ), while ReasonIR-8B-0.5scale uses public data plus 50% randomly sampled VL and HQ data (public+50%VL+50%HQ).**
>
> | | Bio. | Earth. | Econ. | Psy. | Rob. | Stack. | Sus. | Leet. | Pony | AoPS | TheoQ. | TheoT. | Avg. |
> |---|---|---|---|---|---|---|---|---|---|---|---|---|---|
> | Evaluate with original query|
> | ReasonIR-8B | 26.2 | 31.4 | 23.3 | 30.0 | 18.0 | 23.9 | 20.5 | 35.0 | 10.5 | 14.7 | 31.9 | 27.2 | 24.4 |
> | ReasonIR-8B-0.5scale | 19.1 | 25.6 | 19.9 | 26.4 | 18.6 | 24.0 | 17.7 | 33.7 | 6.1 | 12.0 | 25.7 | 20.7 | 20.8 |
> | Evaluate with GPT4 Reason-query|
> | ReasonIR-8B | 43.6 | 42.9 | 32.7 | 38.8 | 20.9 | 25.8 | 27.5 | 31.5 | 19.6 | 7.4 | 33.1 | 35.7 | 29.9 |
> | ReasonIR-8B-0.5scale | 35.5 | 39.1 | 29.0 | 37.0 | 21.4 | 32.0 | 23.9 | 28.4 | 23.7 | 7.1 | 30.3 | 33.0 | 28.4 |

---

> > ### Comment · Reviewer_4Dkf · 2025-06-07
> >
> > Thank you for the additional inputs, they look quite reasonable to me.

---

> > > ### Author Response · Authors · 2025-06-09
> > >
> > > We appreciate the reviewer's thoughtful feedback and are pleased that our responses address the raised concerns. We remain available to provide any additional clarification or discussion that would be helpful!

---

### Author Response · Authors · 2025-06-03

We thank all reviewers for their time and effort in reviewing our submission. During the author response period, we provide additional results, analysis, and explanations to address the comments from the reviewers. We provide a summary below:

**Recognized Contributions**
* (Reviewers 4Dkf, P7d1, MLfg, EF6c) Our retrieval model - ReasonIR-8B, trained using our synthetic data, achieves state-of-the-art performance in reasoning-intensive information retrieval and demonstrates effectiveness in retrieval-augmented generation.
* (Reviewers 4Dkf, P7d1, MLfg) The ideas of hard query and hard negative generation are interesting and effective.
* (Reviewers P7d1, EF6c) The motivation and insights for synthetic data generation are clear and well-grounded.
* (Reviewers MLfg, EF6c) Our ReasonIR model can better leverage test-time techniques.

**Main Results for Rebuttal**

* **Dataset scaling.** Both reviewers 4Dkf and EF6c suggest studying scaling laws for our synthetic data. We agree that studying the scaling behavior of dataset size and composition is important. Within our computational budget, we prioritized data composition ablations in Table 2 of our submission. However, scaling studies are computationally intensive, making a full scaling study beyond our budget. We briefly discussed scaling studies for future work in Section 7 and will expand within budget. Nonetheless, we provide a preliminary study: subsampling 50% of the data decreased BRIGHT performance from 24.4 to 20.8 with original queries and 29.9 to 28.4 with GPT4 Reason-query. Results indicate that scaling up training data could improve performance, which we defer to future work.

* **Qualitative analysis.** As suggested by reviewers P7d1 and MLfg, we provide qualitative examples to analyze (1) the failure mode of ReasonIR and how it can be compensated by BM25, and (2) the limitation of using GPT4 rewriting, specifically for LeetCode retrieval.

* **Data synthesis with small LLMs.** In response to reviewer P7d1, we study the potential of synthesizing hard-query data using a small LLM (LLama3.1-8B in case), and demonstrate that these data are of much lower difficulty and hence less valuable compared to the ones generated using large LLMs used in our paper.

* **Additional comparison with Rank1.** As suggested by reviewer MLfg, we additionally compared ReasonIR with rank1 (with BM25 candidates) on (1) reasoning retrieval in BRIGHT with the original query and (2) non-reasoning retrieval with the subset of BEIR used by rank1. We show that ReasonIR outperforms rank1 in both scenarios.

* **Potential knowledge leakage.** In response to reviewer MLfg, we explain our reasons for why using the documents from the datastore is unlikely to cause knowledge leakage, particularly because we do not use any knowledge of the test data during document selection and query generation. We also recall that knowing the document distribution in the datastore does not necessarily bring performance gains, as reflected by the results of training with easy-query (EQ) data shown in Table 2 of the main paper.

* **Additional evaluation on BEIR.** As suggested by reviewer EF6c, we add evaluations of models trained using different compositions of ReasonIR data on certain BEIR retrieval tasks. The results show that ReasonIR data are effective in improving non-reasoning retrieval tasks compared to training only with the public data.

---

### Decision · Program_Chairs · 2025-07-08

**Decision:**

Accept

**Comment:**

The authors present ReasonIR-8B targeting reasoning-intensive information retrieval tasks. The propose a synthetic data generation pipeline that creates two types of training data: varied-length queries and hard queries  to improve reasoning performance - and thereby helping them achieve SOTA performance on a benchmark and also demonstrate effectiveness over dense retrievers, reranking approaches in downstream RAG tasks.

With respect to the discussion, reviewers raised important questions (i) about the quality of the synthetic data and scaling — and the authors were able to provide satisfactory preliminary scaling studies and additional experiments for data quality (ii) about the OOD capabilities and generalization — which again the authors were able to sufficiently address (iii) comparison with rank 1 - which was again addressed.

Overall, the paper makes solid contributions to an important problem, and while some limitations remain (full scaling studies), the work establishes a solid foundation for future research in reasoning intensive retrieval and RAG